# Expression of intron-containing HIV-1 RNA induces NLRP1 inflammasome activation in myeloid cells

Sallieu Jalloh[1], Ivy K. Hughes[1], Hisashi Akiyama[1], Aldana D. Gojanovich[2], Andres A. Quiñones-Molina[1], Mengwei Yang[2], Andrew J. Henderson[1,3], Gustavo Mostoslavsky[1,2], Suryaram Gummuluru[1]*

1 Department of Virology, Immunology & Microbiology, Boston University Chobanian & Avedisian School of Medicine, Boston, Massachusetts, United States of America, 2 Center for Regenerative Medicine (CReM), Boston University Chobanian & Avedisian School of Medicine and Boston Medical Center, Boston, Massachusetts, United States of America, 3 Department of Medicine, Section of Infectious Diseases, Boston Medical Center, Boston, Massachusetts, United States of America

* rgummulu@bu.edu

## Abstract

Despite the success of antiretroviral therapy in suppressing plasma viremia in people living with human immunodeficiency virus type-1 (HIV-1), persistent viral RNA expression in tissue reservoirs is observed and can contribute to HIV-1-induced immunopathology and comorbidities. Infection of long-lived innate immune cells, such as tissue-resident macrophages and microglia may contribute to persistent viral RNA production and chronic inflammation. We recently reported that de novo cytoplasmic expression of HIV-1 intron-containing RNA (icRNA) in macrophages and microglia leads to MDA5 and MAVS-dependent innate immune sensing and induction of type I IFN responses, demonstrating that HIV icRNA is a pathogen-associated molecular pattern (PAMP). In this report, we show that cytoplasmic expression of HIV-1 icRNA also induces NLRP1 inflammasome activation and IL-1β secretion in macrophages and microglia in an RLR- and endosomal TLR-independent manner. Infection of both macrophages and microglia with either replication-competent or single-cycle HIV-1 induced IL-1β secretion, which was attenuated when cytoplasmic expression of viral icRNA was prevented. While IL-1β secretion was blocked by treatment with caspase-1 inhibitors or knockdown of NLRP1 or caspase-1 expression in HIV-infected macrophages, overexpression of NLRP1 significantly enhanced IL-1β secretion in an HIV-icRNA-dependent manner. Immunoprecipitation analysis revealed interaction of HIV-1 icRNA, but not multiply-spliced HIV-1 RNA, with NLRP1, suggesting that HIV-1 icRNA sensing by NLRP1 is sufficient to trigger inflammasome activation. Together, these findings reveal a pathway of NLRP1 inflammasome activation induced by de novo expressed HIV icRNA in HIV-infected myeloid cells.

**Data availability statement:** All relevant data are within the paper and its Supporting information files.

**Funding:** This work was supported by NIH grants R01DA051889 (to SG and GM), R01AG060890 (to SG) R01DA059952 (to SG and AH), R01AI187175 (to SG and AH), R01DA055488 (to SG and AH), and P30AI042853 (SG, AH and HA). The funders had no role in study design, data collection and analysis, decision to publish, or preparation of the manuscript.

**Competing interests:** The authors have declared that no competing interests exist.

**Abbreviations:** CARD, caspase-1 activation and recruitment domain; CSF, cerebrospinal fluid; CTE, constitutive transport element; dNs, deoxynucleosides; FIIND, function-to-find-domain; gRNA, genomic RNA; HAND, HIV-associated neurocognitive disorders; HIV-1, human immunodeficiency virus type-1; icRNA, intron-containing RNA; iMGs, iPSC-derived microglia; LPS, lipopolysaccharide; LRR, leucine-rich repeat; MDMs, monocyte-derived macrophages; MPMV, Mason–Pfizer monkey virus; msRNA, multiply spliced viral RNA; NLRs, nucleotide-binding leucine-rich repeats; NNRTI, non-nucleoside RT inhibitor; PAMPs, pathogen-associated molecular patterns; PFA, paraformaldehyde; PWH, people living with HIV-1.

## Introduction

Dysfunctional and persistent innate immune activation in people living with human immunodeficiency virus type-1 (HIV-1) people living with HIV-1 (PWH) has been linked to several pathologies generally referred to as HIV-associated non-AIDS complications, such as atherosclerosis, cardiovascular disease, cancers, and HIV-associated neurocognitive disorders (HAND) [1–5]. Long-lived HIV-1-infected cells, such as tissue-resident macrophages, including CNS-resident microglia harboring transcriptionally competent proviruses, can persistently express viral transcripts [6–9]. Importantly, persistent expression of HIV RNA in tissue reservoirs and cerebrospinal fluid (CSF) [10–12] has been associated with the chronic inflammatory state in PWH on antiretroviral therapy [13–16].

Inflammasomes are multiprotein complexes activated by pathogens [17,18] via direct detection of pathogen-associated molecular patterns (PAMPs), or indirect sensing of pathogen-encoded enzymes or perturbations to cellular homeostasis. The ultimate outcomes of inflammasome activation are proinflammatory cytokine (IL-1β and IL-18) secretion and pyroptosis [17,19–25]. Distinct inflammasome receptors such as members of the nucleotide-binding oligomerization domain, and nucleotide-binding leucine-rich repeats (NLRs), play important roles in innate immune response to microbial infections by promoting cell death of infected cells and secretion of IL-1β and IL-18 in response to pathogen stressors [18,19,26–28]. Upon detection of pathogen-derived molecules, nucleation of the NLRs containing the N-terminal pyrin domain, such as NLRP1 and NLRP3, results in recruitment and autoproteolytic processing of proinflammatory caspases caspase-1 and -4 [17,29,30]. Subsequent cleavage of pro-IL-1β, pro-IL-18, and membrane-associated gasdermin D (GSDMD) by activated caspase-1/4 drives the extracellular release of mature IL-1β and IL-18 and membrane pore formation by the N-terminal domain of GSDMD resulting in pyroptotic cell death [31–33].

Chronic HIV-1 infection has been linked to systemic inflammation [34,35]. While elevated levels of IL-1β and IL-18 have been observed in PWH [36–39], the identity of the PAMPs and mechanism(s) for inflammasome activation remain unclear. For instance, transient transfection of HIV-1 genomic RNA (gRNA) or accumulation of reverse transcription intermediates in abortively infected cells have been shown to prime and activate NLRP3 and IFI16 inflammasomes, respectively [40–44]. Additionally, overexpression of recombinant HIV-1 proteins such as gp120, viral protein R (Vpr), viral protein U (Vpu), and transactivator of transcription (Tat) can activate NLRP3 inflammasome and IL-1β production in myeloid cells [22,45–51]. Interestingly, it was recently reported that HIV-1 protease (PR) activity can trigger CARD8 inflammasome activation in macrophages and CD4 + T cells [52–54]. Furthermore, recent studies by our group and others have shown that expression and nuclear export of HIV-1 intron-containing RNA (icRNA) in monocyte-derived macrophages (MDMs), monocyte-derived dendritic cells, and iPSC-derived microglia (iMGs) induced MDA5 and MAVS-dependent type I IFN responses [55–59].

In this study, we demonstrate that establishing productive HIV-1 infection and cytosolic viral icRNA expression activates NLRP1 inflammasome and IL-1β secretion in

macrophages and iMGs. Inhibition of early steps in the viral life cycle that precede proviral transcription, or nuclear export of HIV-1 icRNAs abrogated inflammasome activation. Intriguingly, knockdown of NLRP1 alone, but not NLRP3, AIM2 or CARD8, attenuated HIV-1 icRNA-induced IL-1β secretion. Mechanistically, HIV-1 icRNA-induced inflammasome activation did not require cytosolic RLR/MAVS or endosomal TLR-mediated RNA sensing pathways. Rather, HIV-1 icRNA was recognized by NLRP1, and this interaction promoted caspase 1 activation and IL-1β secretion. Collectively, these findings suggest that HIV-1 icRNA in the cytosol is a unique PAMP, which leads to innate immune activation by two distinct mechanisms in myeloid cells.

## Results

### De novo expression of HIV-1 intron-containing RNA (icRNA) induces caspase-dependent IL-1β activation in macrophages

To identify the potential triggers of diverse innate immune responses in macrophages, we differentiated peripheral blood CD14+ monocytes into macrophages (MDMs), and infected these cells with replication-competent, CCR5-tropic HIV-1 (herein referred to Lai/YU-2env) together with SIVmac239 Vpx-incorporating VLPs (Vpx-VLPs) in the absence or presence of RT (EFV), integrase (Ral), transcription (spironolactone) or CRM1 (KPT335, blocks Rev-mediated HIV icRNA export and Gag expression) inhibitors. Establishment of productive HIV infection and secretion of cytokines was measured by ELISA. We previously showed that HIV-1 infection in MDMs resulted in IP-10 secretion [56]. Here, we show that HIV-1 infection of MDMs also leads to secretion of IL-1β, which was abrogated upon pre-treatment of cells with HIV-1 RT, integrase, transcription, or Rev-CRM1 inhibitors (Fig 1A and 1B), suggesting that nucleo-cytoplasmic export of unspliced or partially-spliced HIV-1 RNA is required for both type I IFN responses and IL-1β secretion. Consistent with the requirement for inflammatory caspase activity for processing of pro-IL-1β into mature IL-1β [60], we observed that pre-treatment of cells with either a pan-caspase (zVAD-fmk) or a caspase-1/caspase-4 inhibitor (VX-765) markedly abrogated HIV-1-induced IL-1β secretion in primary MDMs (Fig 1C and 1D).

Since human monocytes and macrophages have been hypothesized to require priming for inflammasome activation and IL-1β production, we also included TNFα for MDM stimulation, even though HIV-1 icRNA-mediated inflammasome activation in primary MDMs infected with Lai/YU-2env or VSV-G pseudotyped single-cycle HIV-1 did not require priming by exogenous cytokines or other stimuli (Fig 1). Priming with TNFα led to slightly enhanced infection levels and IL-1β secretion in the absence of Vpx-VLPs (S1A and S1B Fig). Supplementation with Vpx-VLPs not only enhanced infection by over 100-fold, but was also sufficient to trigger IL-1β secretion even in the absence of TNFα (S1A and S1B Fig). Since, SAMHD1 inactivation or exogenous Vpx expression in MDMs has been hypothesized to induce innate immune responses [61,62], primary MDMs were infected with Lai/YU-2env in the presence of deoxynucleosides (dNs) to overcome RT restriction in MDMs [63]. Pretreatment with exogenous dNs enhanced HIV-1 infection in MDMs to levels similar to that observed upon pre-treatment with Vpx-VLPs (S1C Fig) and resulted in robust secretion of IL-1β, (S1D Fig), suggesting that HIV-1 infection and viral icRNA expression alone in the absence of priming stimuli can activate inflammasomes in MDMs. To further confirm the requirement of HIV-1 icRNA nuclear export for IL-1β secretion, we infected primary MDMs with VSV-G-pseudotyped single-round HIV-1 expressing GFP (LaiΔenvGFP/G, MOI = 1) wildtype (WT) or a rev-deficient mutant (M10) (Fig 1G). As described in our previously published studies [55,56], HIV-1/M10 infection of MDMs selectively attenuates nucleocytoplasmic export of HIV-1 icRNA while preserving multiply spliced viral RNA (msRNA) nuclear export and GFP reporter expression (Fig 1G). In contrast to WT HIV-1 infection of MDMs, caspase-1 activation (Fig 1H) and IL-1β secretion (Fig 1I) were not observed in HIV-1/M10-infected MDMs (Fig 1I). Collectively these findings suggest that cytoplasmic expression of HIV-1 icRNAs is required for inflammasome activation in HIV-1-infected primary macrophages.

### HIV-1 icRNA-induced inflammasome activation in macrophages is protease-independent

Recent studies have demonstrated HIV-1 PR-dependent CARD8 inflammasome activation in productively infected cells, which was dependent either on transient overexpression of Gag-Pol polyprotein or non-nucleoside RT inhibitor

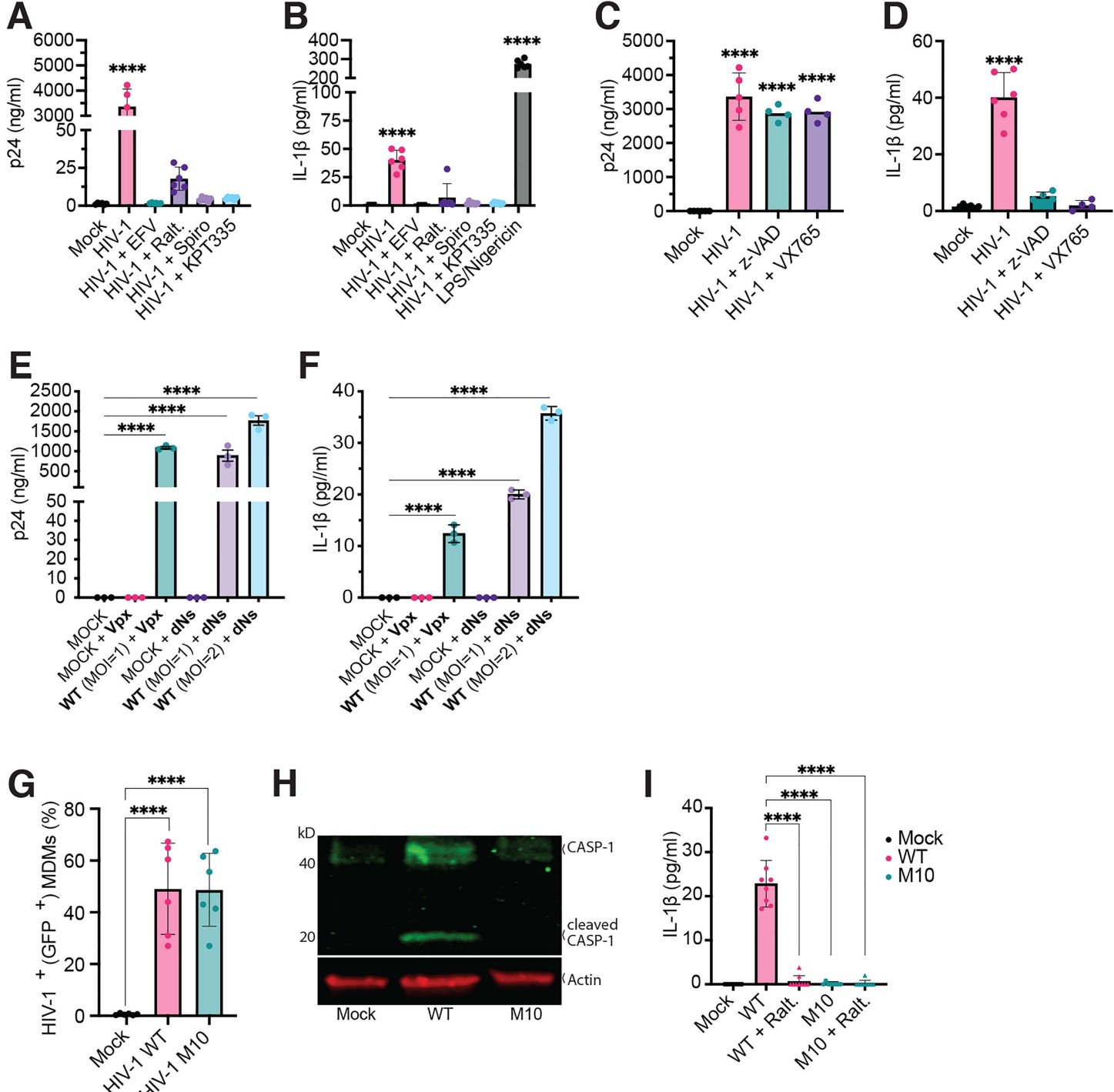

**Fig 1. De novo expression of HIV-1 icRNA induces inflammasome activation in macrophages. (A and B)** MDMs were infected with HIV-1 Lai/YU-2env (MOI 1 in the presence of Vpx VLPs) and production of (A) p24$^{Gag}$ or **(B)** IL-1β in the culture supernatants at 3 days post-infection (dpi) was quantified by ELISA. Infections were done in the absence or presence of HIV-1 inhibitors: EFV: efavirenz (1 μM), Ralt: raltegravir (30 μM), Spiro: spironolactone (100 nM), KPT335: verdinexor (1 μM). Alternatively, MDMs were stimulated with LPS (0.25 μg/ml, 4 h), followed by Nigericin (15 nM, 30 min). **(C and D)** MDMs were infected with Lai/YU-2env (MOI 1, in the presence of pan-caspase (z-VADfmk, 100 nM) or caspase-1/4 specific inhibitor (VX765, 1μM)), and **(C)** p24$^{Gag}$ production or **(D)** IL-1β secretion was measured by ELISA in the culture supernatants at 3 dpi. **(E and F)** MDMs infected with Lai/

YU-2env (MOI 1 or 2) in the presence of either Vpx VLPs or dNs, and (E) p24$^{Gag}$ production or (F) IL-1β secretion in the culture supernatants was quantified at 3 dpi by ELISA. (G–I) Infection of MDMs with LaiΔenvGFP/G (MOI 1) WT or M10. (G) % infection (GFP⁺) at 3 dpi was quantified by flow cytometry. (H) Cleaved caspase-1 and β-actin expression in the whole cell lysates was analyzed by immunoblotting. (I) IL-1β secretion by MDMs infected with HIV-1 (WT or M10) at MOI 1 in the absence or presence of integrase inhibitor (raltegravir, 30 μM), was measured by ELISA. The means ± SEM are shown, and each symbol represents an independent donor. Statistical significance was determined by one-way ANOVA followed by Dunnett's post-test, compared to the uninfected (mock) control. P-values: ****<0.0001; no symbol: not significant (p ≥ 0.05). The data underlying this figure can be found in S1 Data and S1 Raw Images.

(NNRTI)-induced Gag-Pol dimerization in infected cells [53,54,64]. Since HIV-1 unspliced RNA encodes for the Gag-Pol polyprotein, including the viral PR, we sought to determine whether Gag-Pol polyprotein expression and viral PR activity were essential for inflammasome activation and IL-1β secretion in HIV-1-infected MDMs. We generated an HIV-1 provirus in which the *gag-pol* sequence was deleted (HIV-1/ΔGag-Pol, [56]), rendering it null for HIV-1 Gag and PR expression. Both Gag-Pol polyprotein and immature p55$^{Gag}$ expression were not observed in HEK293T cells transfected with HIV-1/ΔGag-Pol proviral plasmid (Fig 2A). We confirmed similar levels of GFP expression upon infection with HIV-1/ΔGag-Pol to that observed with HIV-1/WT and HIV-1/M10 in primary MDMs at 3 days post-infection (dpi) (Fig 2B). Strikingly, IL-1β secretion was detected in MDMs infected with HIV-1 lacking PR activity or Gag-Pol expression (HIV-1/ΔGag-Pol) in an MOI-dependent manner (S1C and S1D Fig) and IL-1β was significantly higher than that of HIV-1/M10 or uninfected

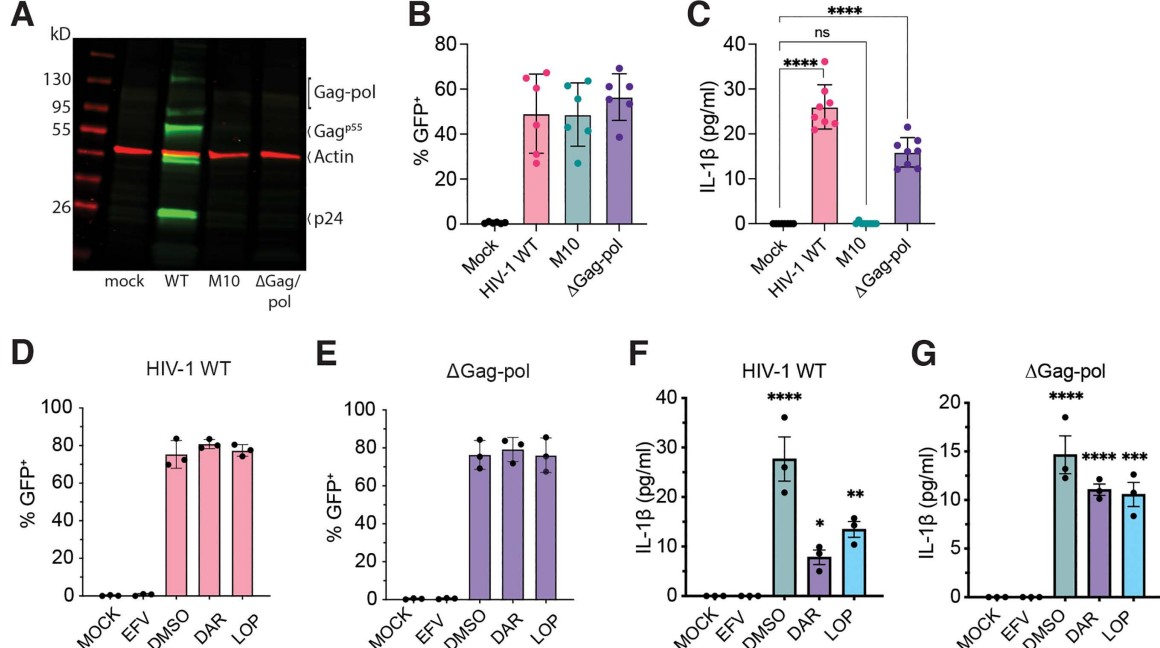

**Fig 2. HIV-1 icRNA-induced inflammasome activation in macrophages is protease-independent.** (A) Immunoblotting analysis of HEK293T cell lysates transfected with WT, M10, or ΔGag-pol HIV-1 constructs, compared to mock (no HIV-1 plasmid transfection). (B) % infection (GFP⁺) in MDMs (MOI 1) was measured by flow cytometry at 3 dpi. (C) Secretion of IL-1β in the culture supernatants of HIV-infected MDMs at 3 dpi measured by ELISA. Infection of MDMs with WT (D) or ΔGag-pol (E) HIV-1 (MOI 1, 3 dpi) in the presence of HIV-1 protease (PR) inhibitors, Darunavir (DAR, 1μM) and Lopinavir (LOP, 1 μM), or RT inhibitor (EFV, 1 μM), compared to untreated (DMSO) control. % infection (GFP⁺) quantified by flow cytometry. IL-1β secreted in the culture supernatants from WT (F) or ΔGag-pol (G) infected MDMs in the absence (DMSO) or presence of DAR, LOP, or EFV at 3 dpi, quantified by ELISA. The means ± SEM are shown, and each symbol represents an independent donor. Statistical significance was determined by one-way ANOVA followed by Dunnett's post-test (C, F, G), compared to mock. P-values: ****<0.0001; ***<0.001; **<0.01; *<0.05; no symbol: not significant (p ≥ 0.05). The data underlying this figure can be found in S2 Data.

(mock) MDMs, although somewhat attenuated compared to HIV-1/WT (Fig 2C), suggesting that PR activity was dispensable for HIV-1 icRNA-induced inflammasome activation in MDMs.

To confirm this HIV-1 PR-independent mechanism of inflammasome activation, we infected MDMs with either WT or PR-deficient HIV-1 (ΔGag-pol, (MOI = 1) for 3 days in the absence (DMSO) or presence of HIV-1 PR inhibitors (Darunavir (DAR) and Lopinavir (LOP)), or an HIV-1 RT inhibitor (efavirenz (EFV)) (Fig 2D). While HIV-1 infection establishment was unaffected upon treatment with PR inhibitors (Fig 2D and 2E), both DAR and LOP pretreatment reduced, but did not completely abrogate IL-1β secretion in HIV-1/WT-infected MDMs (Fig 2F), while having a negligible impact on IL-1β secretion in HIV-1/ΔGag-Pol-infected MDMs (Fig 2G, respectively). The lack of complete attenuation of IL-1β secretion in HIV-1/WT-infected MDMs in the presence of PR inhibitors suggests that cytoplasmic HIV-1 icRNA expression alone triggers inflammasome activation and IL-1β secretion in MDMs, independent of HIV-1 PR activity or Gag-Pol dimerization.

### Cytoplasmic expression of de novo HIV-1 icRNA in iMGs leads to inflammasome activation

Since we recently reported that human iMGs) are highly susceptible to HIV-1 infection [55], we next sought to look at the ability of HIV-1 icRNA to induce IL-1β release in iMGs. We generated iMGs from hematopoietic progenitors derived from two iPSC lines (Fig 3A), and obtained a highly pure microglia population (>98% CD45+CD11b+), expressing microglia-specific markers TREM2 and P2RY12 (Fig 3B). While HIV-1 infection of myeloid cells is antagonized by SAMHD1 [61,65–69], iMGs expressed lower levels of antiviral (dephosphorylated) SAMHD1 compared to primary MDMs (S2A Fig) and were permissive to both Lai/YU-2env (Figs 3C and S2B) and LaiΔenvGFP/G (S2C Fig), and did not require co-infection with Vpx-VLPs or supplementation with excess dNs to boost infection. Importantly, infection of iMGs from both iPSC lines (iMG #1 and iMG #2) with Lai/YU-2env resulted in an MOI-dependent increase in p24$^{Gag}$ secretion (Fig 3C; left panel) and IL-1β secretion (Fig 3C; right panel) at day 3 pi, which was suppressed upon EFV treatment. Infection of iMGs with single-cycle HIV-1 (LaiΔenvGFP/G) also resulted in robust infection (S2D Fig, left panel) and led to significant IL-1β production which was abrogated upon EFV pretreatment (S2D Fig, right panel), suggesting that viral spread was not required for IL-1β secretion. These results suggest that similar to the findings in MDMs, HIV-1 infection in primary microglia leads to inflammasome activation and IL-1β release.

We next sought to determine whether cytoplasmic HIV-1 icRNA expression alone, independent of Gag-Pol expression or viral PR activity, is a trigger of inflammasome activation and IL-1β secretion in iMGs, similar to the findings in MDMs (Fig 2). To this end, we infected iMGs with single-cycle HIV-1 WT or mutant (M10 and ΔGag-pol) viruses (MOI = 1) and confirmed infection establishment by RT-qPCR analysis for viral RNA expression from cytoplasmic fractions of infected cells at 3 dpi (Fig 3D and 3E). While similar levels of multiply-spliced viral transcripts (tat/rev) were observed in microglia infected with either WT, M10, or ΔGag-Pol viruses (Fig 3D), unspliced viral RNA (icRNA transcripts) were only detected in cytoplasmic fractions of WT or ΔGag-Pol, but not in M10-infected microglia (Fig 3E). In correlation to cytoplasmic viral icRNA expression, infection of iMGs with either WT or ΔGag-Pol/HIV-1, but not M10, resulted in IL-1β secretion (Fig 3F). These findings suggest that inflammasome activation in microglia requires cytoplasmic viral icRNA expression and that, similar to primary MDMs (Figs 1 and 2), Gag-Pol expression or PR activity is not essential for IL-1β secretion.

### Cytoplasmic expression of HIV-1 icRNA induces inflammasome activation in macrophages in a MAVS-independent manner

Previously published findings by us and others have demonstrated the selective requirement of the Rev/CRM1-mediated nuclear export of HIV-1 icRNA for inducing MDA5-dependent type I responses [57,59]. In contrast, HIV-1 icRNA nuclear export mediated by the constitutive transport element (CTE) from Mason–Pfizer monkey virus (MPMV) failed to induce type-I interferon responses [56]. Hence, we sought to confirm the requirement of Rev-CRM1 nuclear export pathway in triggering icRNA-mediated inflammasome activation and IL1-β secretion. MDMs were infected with single-cycle VSV-G pseudotyped WT LaiΔenvGFP/G (Rev/CRM1-dependent nuclear export of HIV-1 icRNA), M10 (Rev-deficient), or M10-4xCTE virus (MPMV CTE-dependent nuclear export of HIV-1 icRNA) in the absence (DMSO) or presence of EFV (1 μM).

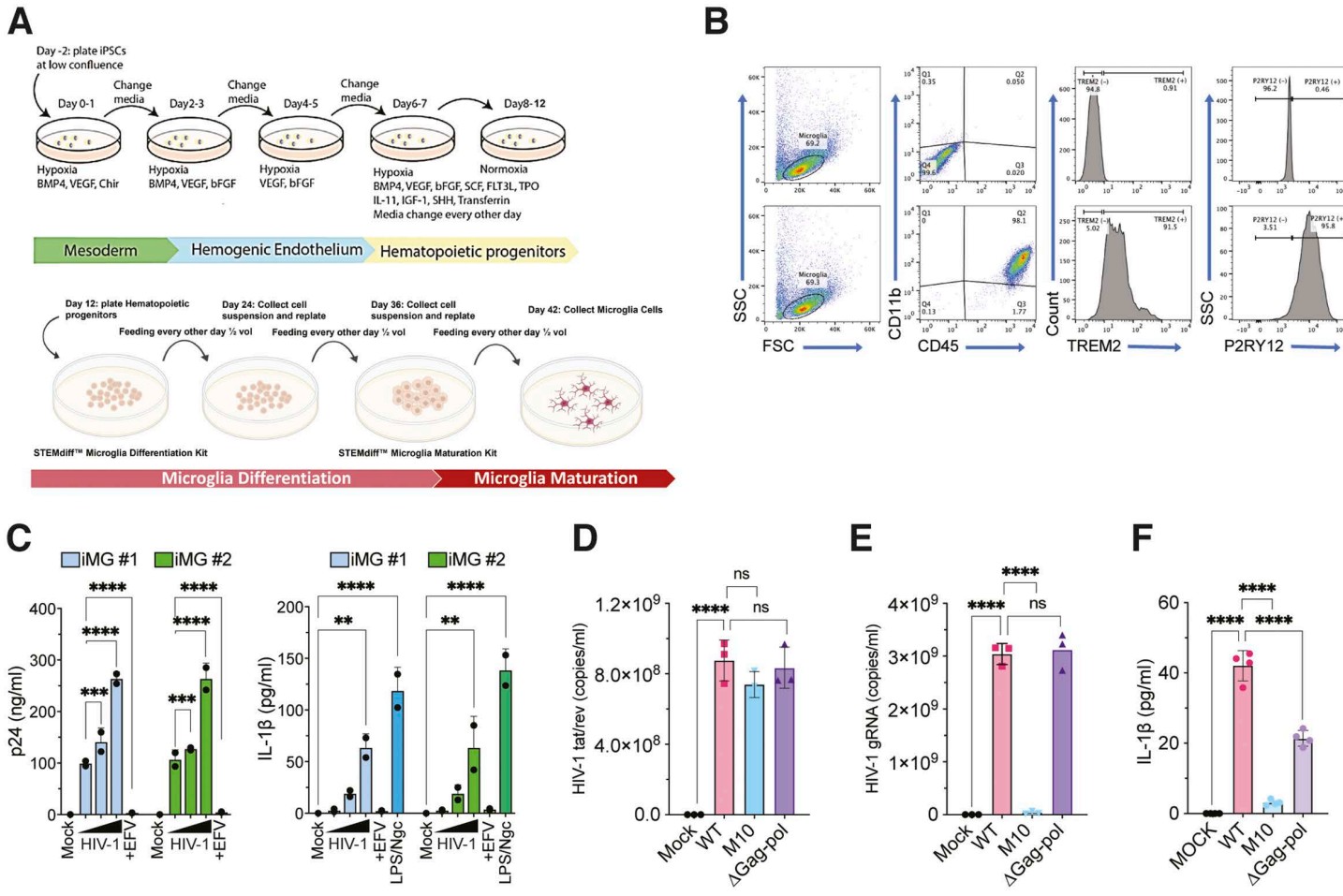

**Fig 3. Cytoplasmic expression of HIV-1 icRNA in iMGs induces inflammasome activation. (A)** Schematic of iMGs differentiation protocol. **(B)** Flow cytometry profiles of iMGs showing expression of myeloid cell markers (CD45 and CD11b) and microglia-specific markers (TREM2 and P2RY12) at day 28 of differentiation. Top panels represent isotype IgG control staining, and bottom panels represent specific antibodies against indicated cell-specific markers. **(C)** iMGs were infected with HIV-1 Lai/YU-2env (MOI 0.1, 0.3, or 1) ± EFV (1 μM). Alternatively, iMGs were stimulated with LPS (0.25 μg/ml, 4 h) followed by Nigericin (15 nm, 30 min). Cell culture supernatants were harvested at 3 dpi for quantifying p24$^{Gag}$ (left panel) and IL-1β (right panel) release by ELISA. RT-qPCR analysis of multiply-spliced (tat/rev) viral transcripts **(D)** and unspliced (usRNA) transcripts **(E)** in iMGs infected with HIV-1 WT, M10, or ΔGag-pol at 3 dpi. The CT values were normalized first to that of GAPDH, and then to uninfected (mock) control. **(F)** Quantitation of IL-1β secretion in the culture supernatants of HIV-infected iMGs at 3 dpi, compared to uninfected (mock) control. The means ± SD are shown, and each symbol represents an independent experiment. Statistical significance was determined by two-way ANOVA followed by the Dunnett's post-test comparing to EFV-treated (C, left panel) or mock (C, right panel); or one-way ANOVA followed by the Dunnett's post-test comparing to HIV-1 WT **(D–F)**. *P*-values: ****<0.0001; ***<0.001; **<0.01; ns: not significant (*p* ≥ 0.05). The data underlying this figure can be found in S3 Data.

Analysis for infection efficiency (%GFP positive) by flow cytometry revealed that GFP expression in productively infected MDMs was equivalent amongst all three viruses (Fig 4A). Notably, the insertion of four copies of CTE from MPMV in the *pol* open reading frame of the M10 mutant virus led to rescue of cytoplasmic p55$^{Gag}$ protein expression (Fig 4B), and unspliced (gRNA) HIV-1 RNA expression (Fig 4C) and spliced (Tat/Rev) transcripts (Fig 4D) in macrophages. In alignment with previously published studies, type-I IFN production was only observed when cytoplasmic expression of HIV-1 icRNA was facilitated by Rev-CRM1 nuclear export pathway (Fig 4E, [56,57]). In contrast, nuclear export and cytoplasmic expression of HIV-1 icRNA by either Rev-CRM1 or MPMV CTE induced IL-1β secretion (Fig 4F). While IL-1β production in HIV-1/

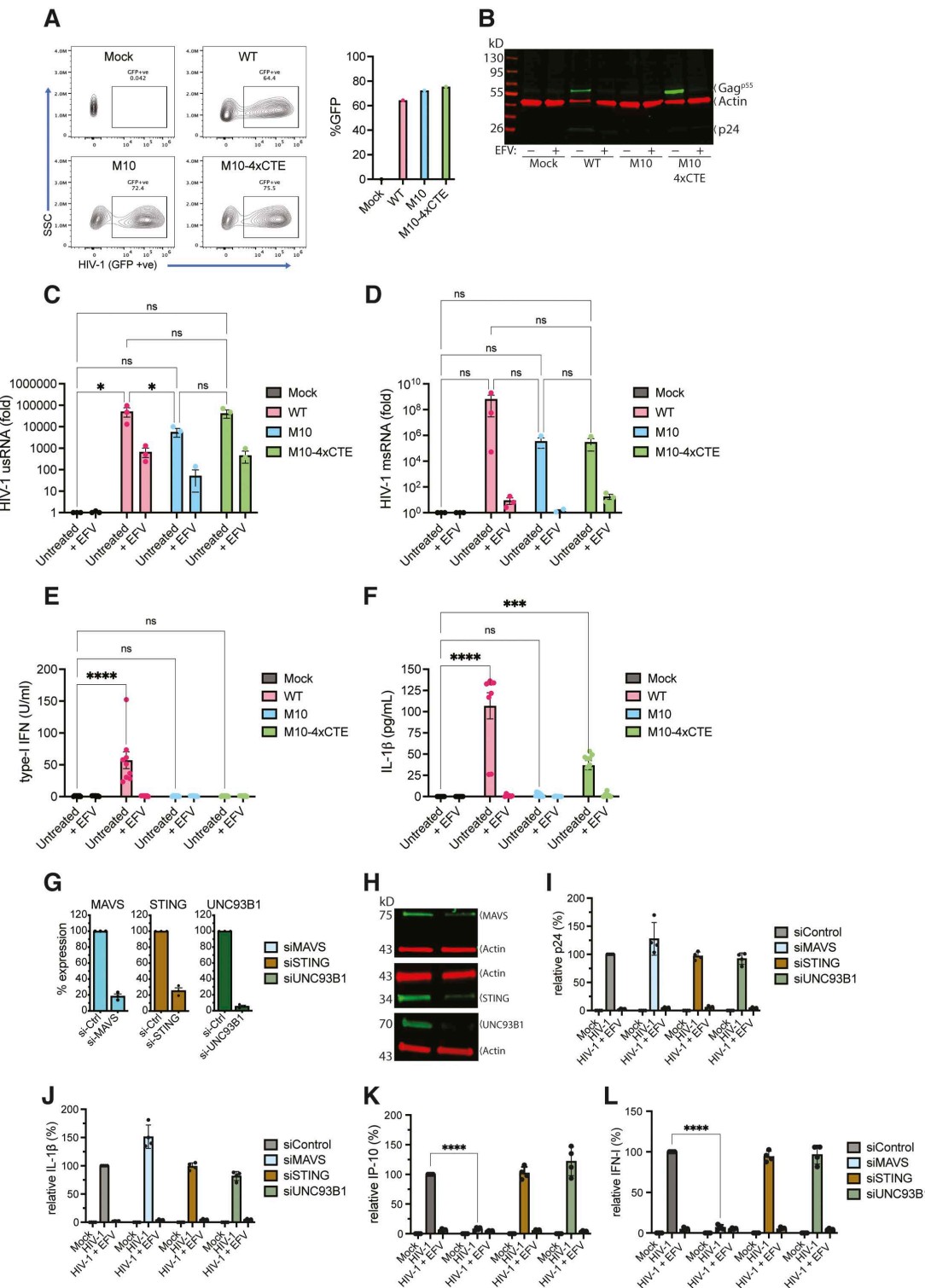

**Fig 4. Cytoplasmic expression of HIV-1 icRNA induces inflammasome activation in macrophages in a MAVS-independent manner. (A)** Flow cytometry analysis (left panel) of MDMs infected with WT or mutant (M10 or M10-4xCTE) LaiΔenvGFP/G viruses (MOI 1) in the presence of SIV3+/Vpx VLPs, compared to mock (no virus infection). % infection (GFP⁺) in MDMs was quantified at 3 dpi (right panel). **(B)** Immunoblotting analysis of whole cell lysates from MDMs infected with HIV-1 (WT, M10, or M10-4xCTE constructs) in the absence (untreated, DMSO) or presence of RT inhibitor (EFV, 1 μM) at 3 dpi. HIV-1 proteins (p55$^{Gag}$ and p24$^{Gag}$) were visualized using anti-HIV-1 IgG, β-actin was used as a loading control. **(C)** RT-qPCR quantification

of unspliced (usRNA), and multiply-spliced (Tat/Rev) transcripts (msRNA) **(D)** in whole cell lysates from MDMs infected with WT, M10, or M10-4xCTE virus±EFV, at 3 dpi compared to uninfected (mock) lysates. HIV-1 unspliced and spliced transcripts were quantified by RT-qPCR. Production of bioactive type-I IFN **(E)**, and secretion of IL-1β **(F)** in culture supernatants from MDMs infected with WT, M10, or M10-4xCTE virus±EFV were harvested at 3 dpi and quantified using type I IFN bioassay or ELISA, respectively. **(G, H)** Transient knockdown of MAVS, STING and UNC93B1 by siRNA transfection in primary MDMs. Knockdown efficiency was quantified by RT-qPCR **(G)** at day 2 post-transfection and reported as relative expression normalized to that observed with scramble siRNA (siControl), and protein level analyzed by immunoblotting **(H)**. siRNA-transfected MDMs were infected with LaiΔenvGF-P/G (MOI 1) for 3 days **(I–L)**. Culture supernatants were harvested and analyzed at 3 dpi for p24$^{Gag}$ **(I)**, IL-1β **(J)**, and IP-10 **(K)** secretion by ELISA. **(L)** Bioactive type I IFN in the culture supernatants from infected MDMs at 3 dpi quantified using interferon bioassay. The values were normalized to that of mock-infected control in each donor. The means±SEM are shown, with each symbol representing an independent donor. The means±SEM are shown, and each symbol represents an independent donor. Statistical significance was determined by two-way ANOVA followed by Tukey's multiple comparisons **(C, D)**, or Dunnett's post-test, comparing HIV-infected MDMs to uninfected (mock) control **(E, F)**, or with specific siRNAs to that from siControl **(K, L)**. *P*-values: ****<0.0001; ***<0.001; **<0.01; *<0.05; ns: not significant (*p*≥0.05). The data underlying this figure can be found in S4 Data.

M10-4xCTE infected macrophages were lower than that observed with HIV-1 WT-infected MDMs, levels were significantly enhanced over that observed with HIV-1/M10 or mock-infected MDMs (Fig 4F).

Myeloid cells utilize a wide range of pattern recognition receptors to sense pathogen determinants and initiate antiviral responses. Diverse viral RNA sensors, including RLRs such as RIG-I and MDA5 [70,71], TLRs [72,73], and NLRs [21,23,24,74–80], sense and respond to viral RNAs. Since our previous studies had implicated MAVS-dependent induction of type I IFN and ISG expression upon cytoplasmic HIV-1 icRNA expression [56], we sought to determine whether HIV-1 icRNA-induced IL-1β secretion also required MAVS-initiated signaling cascades. To this end, we used siRNAs to transiently knock down the expression of adaptor proteins for cytosolic viral nucleic acid sensors (MAVS and STING) or a chaperone protein (UNC93B1), which are important for innate immune activation and type I interferon responses upon RLR, cGAS and endosomal TLR activation, respectively [81–86]. Downregulation of MAVS, STING and UNC93B1 expression was confirmed by RT-qPCR (Fig 4G), and immunoblotting (Fig 4H). siRNA-treated MDMs were either left uninfected (mock) or infected with LaiΔenvGFP/G (MOI=1) in the absence (DMSO) or presence of EFV (1 µM), followed by quantification of p24$^{Gag}$ and IL-1β secretion in culture supernatants at 3 dpi (Fig 4I and 4J).

Transient knockdown of MAVS, STING, or UNC93B1 did not affect HIV-1 infection establishment, as quantified by p24$^{Gag}$ production in the culture supernatants, compared to non-targeted (scramble) control (Fig 4I). Surprisingly, knockdown of MAVS, STING, or UNC93B1 in primary MDMs did not abrogate HIV-1 icRNA-induced IL-1β secretion (Fig 4I). In contrast, IP-10 secretion (Fig 4J) or type I IFN induction (Fig 4K) was markedly diminished upon downregulation of MAVS expression but not STING or UNC93B1, in agreement with our previously published results that HIV-1 icRNA induces type I IFN responses in a MAVS-dependent manner [34,55,56]. These data suggest that cytosolic nucleic acid sensing pathways that require MAVS, STING, or UNC93B1 activities are dispensable for HIV-1 icRNA-mediated inflammasome activation and IL-1β secretion. Collectively, these findings suggest that nuclear export and cytoplasmic expression of HIV-1 icRNA can induce IL-1β production in a MAVS-independent manner, thus underlying the existence of divergent nucleic acid sensing pathways of cytosolic HIV-1 icRNA that trigger type I IFN production and inflammasome activation.

## NLRP1 is required for HIV-1 icRNA-dependent inflammasome activation in macrophages

Besides cytosolic RLRs and endosomal TLRs, NLRs and AIM2-like inflammasomes have been hypothesized to sense HIV-1 genomic RNAs, abortive viral transcripts, and dsDNA (reverse transcription intermediates), respectively [41,43,44,87,88]. Therefore, we sought to determine the potential role of these inflammasome sensors in mediating IL-1β production upon cytoplasmic HIV-1 icRNA expression in macrophages. We used siRNAs to transiently knock down the expression of key inflammasome sensors (AIM2, NLRP1, NLRP3, and CARD8) and executioner caspases (Caspase-1 and Caspase-4) in MDMs, and confirmed significant knockdown of the target RNA expression by RT-qPCR (Fig 5A) and protein expression by immunoblotting (Fig 5B), as well as functional assays (S3A and S3B Fig).

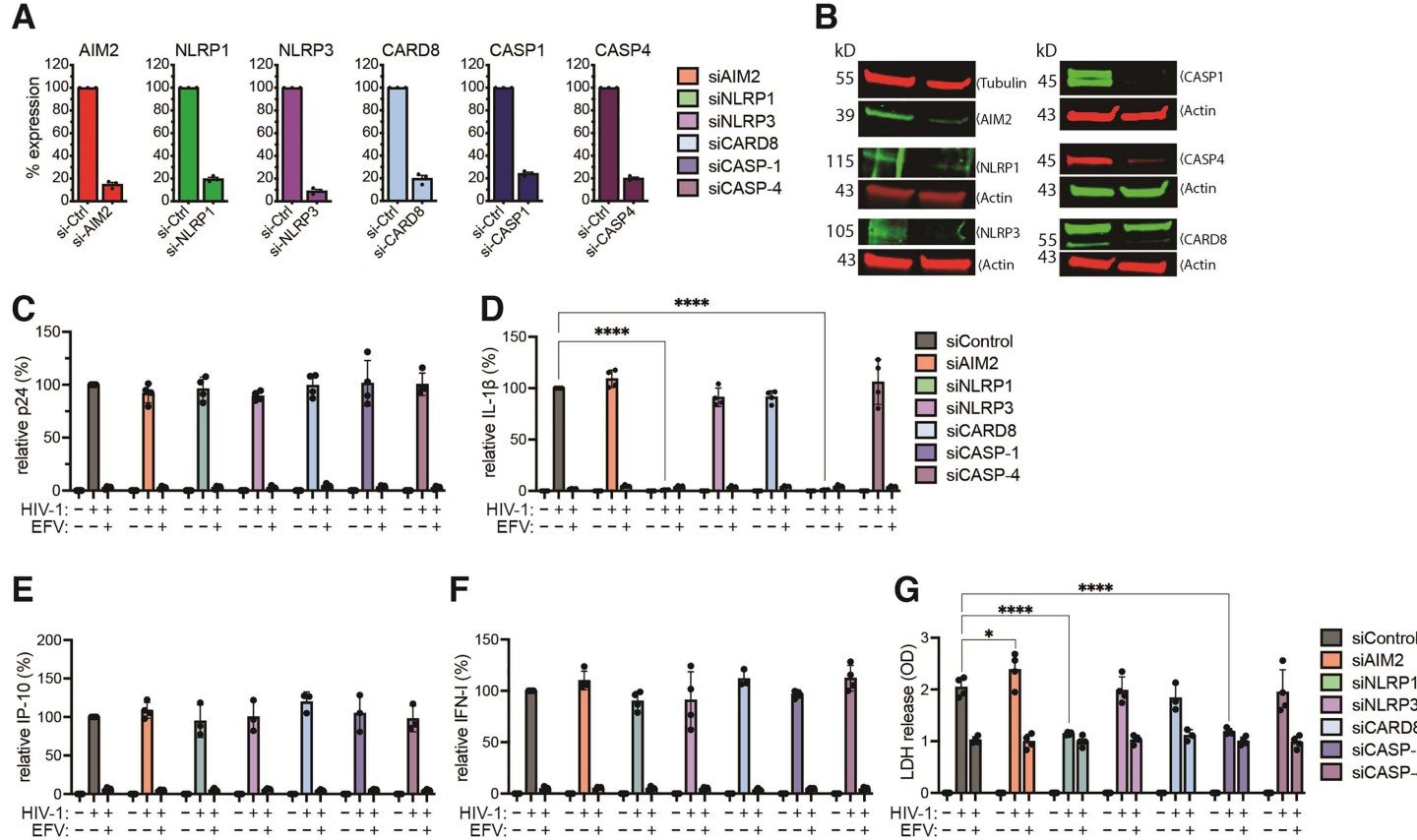

**Fig 5. NLRP1 is required for HIV-1 icRNA-dependent inflammasome activation in macrophages. (A, B)** Transient knockdown of AIM2, NLRP1, NLRP3, CARD8, CASP1, and CASP4 by siRNA transfection in primary MDMs. Knock-down efficiency was determined by RT-qPCR at day 2 post-transfection, and represented as relative mRNA expression of individual targets compared to scramble siControl **(A)**, and protein level analyzed by immunoblotting **(B)**. MDMs were infected with LaiΔenvGFP/G (MOI 1) for 3 days in the absence (DMSO) or presence of EFV **(C–G)**. Culture supernatants were harvested at 3 dpi and analyzed for p24$^{Gag}$ **(C)**, IL-1β **(D)**, and IP-10 **(E)** secretion by ELISA. **(F)** Bioactive type I IFN in the culture supernatants from infected MDMs at 3 dpi quantified using interferon bioassay. The values in panels B-E were normalized to that of siControl-transfected and HIV-1-infected cells in each donor and reported as relative expression. **(G)** Quantitation of LDH release in culture supernatants was measured using the CytoTox cytotoxicity assay, followed by subtraction of background OD from culture media, and normalized to EFV-treatment for each siRNA transduction per donor. The means±SEM are shown, and each symbol represents an independent donor. Statistical significance was determined by two-way ANOVA followed by Dunnett's post-test comparing HIV-infected MDMs from siControl to indicated specific siRNA **(D, G)**. $P$-values: ****<0.0001; *<0.05; no symbol: not significant ($p ≥ 0.05$). The data underlying this figure can be found in S5 Data.

To assess the role of inflammasomes in driving HIV-1 icRNA-induced IL-1β secretion, siRNA-transfected MDMs were either mock-infected or infected with LaiΔenvGFP/G (MOI=1) (Fig 5C and 5D). Notably, the knockdown of inflammasome pathways did not alter HIV-1 infection efficiency in MDMs (Fig 5C). Interestingly, knockdown of NLRP1 and caspase-1 expression, but not AIM2, NLRP3, CARD8, or caspase-4, led to a significant reduction in IL-1β secretion in HIV-1-infected MDMs (Fig 5D). In contrast, knockdown of any of the inflammasome components did not impact HIV-1 icRNA-induced IP-10 (Fig 5E) or type I IFN production (Fig 5F). Furthermore, there was a significant enhancement in LDH release at 3dpi from HIV-1-infected MDMs compared to uninfected (mock) or EFV-treated controls (Fig 5G), highlighting the loss of cell integrity and pyroptotic cell death. Moreover, LDH secretion was abrogated upon knockdown of either NLRP1 or caspase-1, comparable to uninfected or EFV-treated conditions (Fig 5G). Collectively, these findings suggest that NLRP1

inflammasome and caspase-1 activation are required for HIV-1 icRNA-induced IL-1β secretion and pyroptotic cell death in macrophages.

## Overexpression of NLRP1 enables HIV-1 icRNA-dependent IL-1β secretion in THP-1/PMA macrophages

Since the knockdown of NLRP1 in primary MDMs resulted in attenuated IL-1β secretion in HIV-1-infected MDMs (Fig 5), we next sought to determine if NLRP1 expression alone is enough for inflammasome activation and IL-1β release. We over-expressed NLRP1 in THP-1 monocytes, which do not express detectable levels of endogenous NLRP1 (Fig 6A) to determine if NLRP1 expression rescues HIV-1 icRNA-induced IL-1β secretion. THP-1 cells were transduced with a lentivector expressing full-length NLRP1, and NLRP1 overexpression in THP-1 monocytes was confirmed by immunoblotting (Fig 6A). We next differentiated THP-1 monocytes into macrophage-like cells with phorbol 12-myristate-13-acetate (PMA, 100 nM). Immunofluorescence analysis confirmed a lack of NLRP1 expression in parental THP1/PMA macrophages (S4 Fig). In contrast, cytoplasmic NLRP1 expression was detected upon constitutive NLRP1 overexpression in THP1/PMA/ NLRP1 macrophages, similar to MDMs and iMGs (S4 Fig). THP1/PMA macrophages were infected with LaiΔenvGFP/G (MOI = 1) in the absence (infection control) or presence of an RT inhibitor (EFV), integrase inhibitor (Ral), and caspase 1/4 inhibitors (zVAD-fmk and VX-765) in the presence of TNFα. Flow cytometry analysis at 3 dpi showed similar levels of infection (represented by GFP expression) of both control THP1/PMA and THP1/PMA/NLRP1 macrophages (Fig 6B). As expected, infection was blocked in the presence of RT and integrase inhibitors but remained unchanged when cells were pre-treated with caspase inhibitors. Intriguingly, IL-1β secretion was significantly enhanced in HIV-1-infected THP1/ PMA/NLRP1 macrophages compared to control THP1/PMA macrophages (Fig 6C). Additionally, despite robust infection establishment in the presence of both pan-caspase (z-VAD) and caspase-1 (VX-765) inhibitors (Fig 6B), IL-1β production was attenuated in THP1/PMA/NLRP1 macrophages compared to DMSO-treated infection control (Fig 6C). As a positive control, stimulation of both THP1/parental and THP1/PMA/NLRP1 macrophages with a DPP8/9 inhibitor, Val-boroPro (VbP), a known activator of NLRP1 inflammasome [89], induced robust IL-1β secretion only in NLRP1-overexpressing cells, confirming NLRP1 functionality.

To confirm the requirement of cytoplasmic HIV-1 icRNA sensing for NLRP1- inflammasome activation, we infected parental and THP1/PMA/NLRP1 macrophages with LaiΔenvGFP/G WT, M10, or ΔGag-Pol (PR-deficient) viruses (MOI = 1) and observed relatively similar levels of infection between all three viruses when analyzed by flow cytometry (Fig 6D). HIV-1 (WT or ΔGag-pol) infected THP1/PMA/NLRP1 macrophages primed with TNFα, but not TNFα -primed HIV-1/ M10-infected macrophages, secreted robust levels of IL-1β (Fig 6E), thus confirming the requirement of nuclear export of HIV-1 icRNA for PR-independent NLRP1 activation-induced IL-1β secretion. Since IL-1β release only examines the ensemble effects of HIV-1 icRNA-induced inflammasome activation in infected THP1/PMA/NLRP1 macrophages, we sought to examine cell-intrinsic caspase-1 activation in infected cells by flow cytometry. FLICA staining showed robust upregulation of activated caspase-1 in virus-infected THP1/PMA/NLRP1 macrophages (S5A and S5B Fig), which was significantly attenuated upon pre-treatment with RT inhibitor (EFV) (S5B Fig, right panel). Interestingly majority of cells with active caspase-1 were GFP-deficient or uninfected cells (S5A Fig), suggesting that HIV-1 icRNA expression not only triggers caspase-1 activation and infected cell death but also transfer inflammatory responses to bystander cells.

To investigate whether NLRP1-mediated cytosolic HIV-1 icRNA sensing is independent of other cytosolic nucleic acid sensing pathways in myeloid cells, we stably knocked down expression of MAVS or STING in NLRP1[+] THP-1/ PMA cells by lentiviral transduction of shRNAs, and confirmed knockdown of MAVS and STING expression by RT-qPCR (Fig 6F) and immunoblotting (Fig 6G). Compared to control NLRP1[+] THP-1/PMA cells, knockdown of MAVS or STING in NLRP1[+] THP-1/PMA macrophages did not impact infection efficiency (Fig 6H). While knockdown of STING led to slightly lower level of secreted IL-1β compared to control (scramble) cells, knockdown of either adaptors still resulted in robust IL-1β production at 3 dpi compared to uninfected (mock) condition (Fig 6I). Similar to control NLRP1[+] THP-1/PMA cells, pre-treatment with an RT inhibitor (EFV) markedly reduced IL-1β secretion in both cell lines, thus confirming the

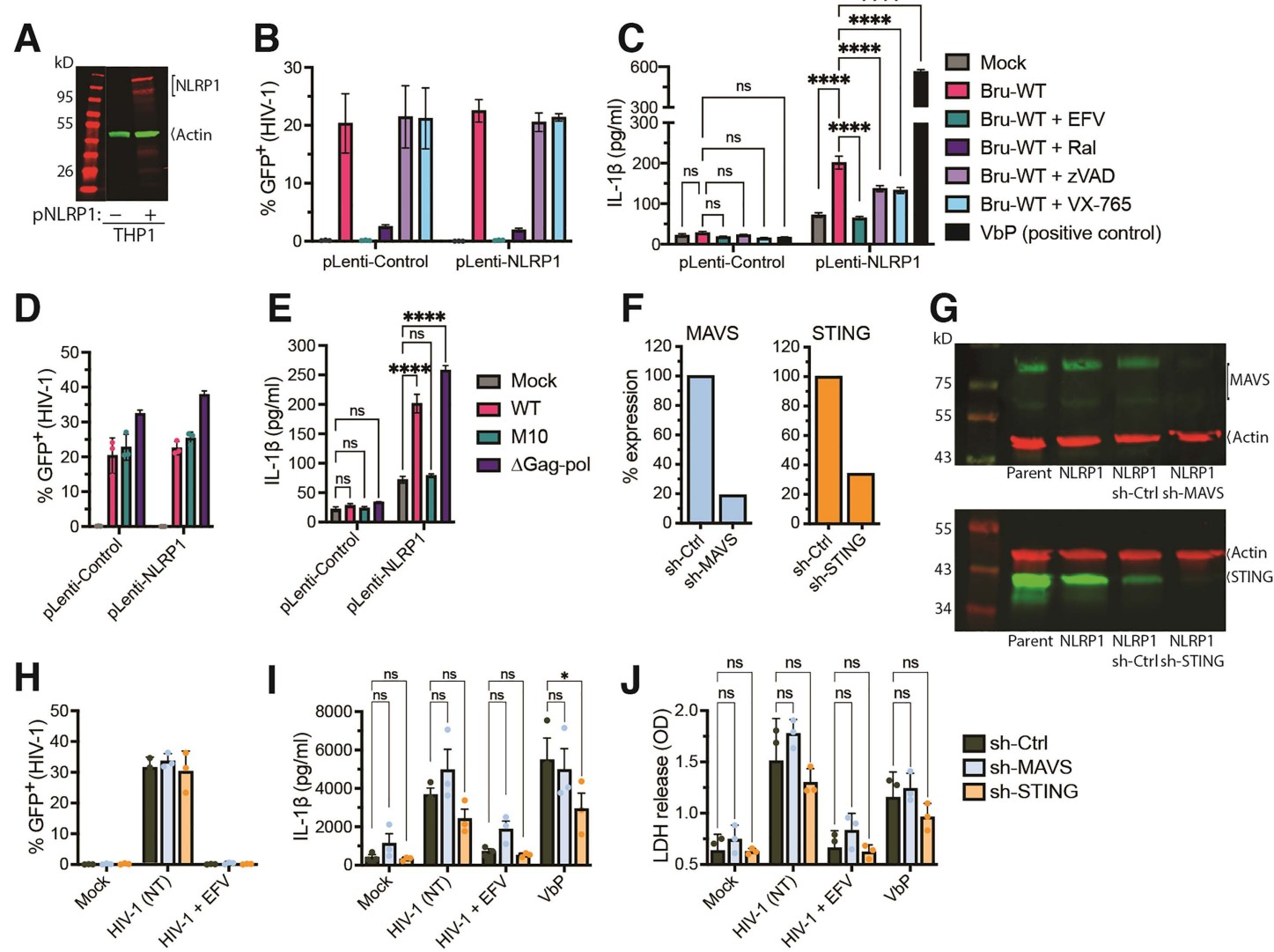

**Fig 6. NLRP1 overexpression in THP-1/PMA macrophages enhances HIV-1 icRNA-induced IL-1 β production.** **(A)** Immunoblotting analysis of NLRP1 expression in THP-1 monocytes transduced with either non-targeting (scramble) control or NLRP1-expressing lentivector. β-actin was used as a loading control. **(B–D)** THP1/PMA macrophages were primed with TNFα (10 ng/ml) and then infected with LaiΔenvGFP/G (MOI 1) in the presence of HIV inhibitors (EFV and Ral) or caspase inhibitors (zVAD-fmk and VX-765). **(B)** % infection (GFP⁺ cells) was quantified by flow cytometry at 3 dpi, and **(C)** IL-1β secretion was analyzed by ELISA. **(D, E)** THP-1/PMA macrophages were primed (TNFα, 10 ng/ml) and then infected with LaiΔenvGFP/G WT, M10, or ΔGag-pol (MOI 1) for 3 days. **(D)** % infection (GFP⁺) was determined by flow cytometry, and **(E)** IL-1β secreted in culture supernatants was measured by ELISA at 3 dpi. Knockdown of MAVS or STING expression in THP1/NLRP1 monocytes transduced with lentiviral vectors expressing scramble shRNA control (scramble), or shRNA sequences against MAVS or STING confirmed by RT-qPCR **(F)** and immunoblotting **(G)**. **(H–J)** Infection of primed (TNFα, 10 ng/ml) THP1/PMA macrophages with LaiΔenvGFP/G (MOI 1) in the absence (no drug treatment, NT) or presence of HIV-1 RT inhibitor (EFV) for 3 days. **(H)** Flow cytometry analysis at 3 dpi showing infection (% GFP expression) in scramble (control) and knockdown cell lines. **(I)** Culture supernatants were harvested at 3 dpi and analyzed for IL-1β production by ELISA. **(J)** Quantitation of LDH release in the culture supernatants at 3 dpi was measured using the CytoTox cytotoxicity assay. The means ± SEM from 3 independent experiments are shown. Statistical significance was determined by two-way ANOVA followed by Dunnett's post-test comparing WT HIV-infected Control or NLRP1-expressing THP-1/PMA macrophages in the absence or presence of indicated inhibitors **(C)** or infections with HIV mutants **(E)**, or WT HIV-infected or VbP-treated THP-1/PMA macrophages expressing scramble shRNA (control) to those expressing specific shRNAs targeting either MAVS or STING **(I–J)**. *P*-values: ****< 0.0001; *< 0.05; no symbol: not significant (*p* ≥ 0.05). The data underlying this figure can be found in S6 Data.

requirement for de novo viral icRNA sensing via NLRP1, independently of MAVS or STING expression. Lastly, quantitation of LDH release in the culture supernatants of HIV-1-infected NLRP1+ THP-1/PMA cells at 3 dpi corroborated the IL-1β data (Fig 6I), with comparable levels of LDH release into the culture supernatants of HIV-1 (WT)-infected control, MAVS or STING knockdown cells (Fig 6J).

Even though CARD8 has been reported to sense HIV-1 protease activity and trigger inflammasome activation in myeloid cells [52,54,90], knockdown of human CARD8 did not abrogate IL-1β secretion in HIV-1-infected primary macrophages (Fig 5C). To further investigate whether protease-dependent CARD8 activation is linked with cytoplasmic HIV-1 icRNA-dependent NLRP1 activation, PMA-differentiated THP-1/parental macrophages were primed with TNFα (10 ng/ml), and then infected with LaiΔenvGFP/G WT or protease-inactive mutant (ΔPR) virus (MOI = 1) for 48 hours, followed by medium change and addition of HIV-1 RT inhibitor rilpivirine (RPV, 5 μM) or DMSO (vehicle control) for 24 hours (S5C–S5E Fig). Cells were harvested and analyzed for infection efficiency based on GFP expression by flow cytometry (S5C Fig), and the culture supernatants were harvested to measure IL-1β production by ELISA (S5D Fig) and LDH release (S5E Fig). Treatment of THP-1/PMA cells at 48 h post-infection with RPV resulted in dramatically increased IL-1β secretion and cell death in cells infected with WT but not ΔPR virus, consistent with the previous reports demonstrating CARD8 cleavage and inflammasome activation following NNRTI-induced Gag-pol dimerization and HIV-1 PR activation [91,92]. To investigate the role of CARD8 in NLRP1-mediated HIV-1 icRNA sensing, we transduced THP-1/control or THP-1/NLRP1 cells with control or CARD8-targeting shRNA (S5F–S5J Fig). Immunoblotting analysis confirmed the successful knockdown of CARD8 (S5F Fig) while having minimal effects on NLRP1 expression (S5G Fig). shRNA-transduced PMA-differentiated THP-1 cells were subsequently primed with TNFα (10 ng/mL) and infected with LaiΔenvGFP/G WT virus (MOI = 1) for 72 h, or treated with VbP (100 μM) for 24 h. Cells and culture supernatants were harvested at the indicated time points, and GFP expression in cells was analyzed by flow cytometry (S5H Fig), IL-1β secretion by ELISA (S5I Fig), and LDH release (S5J Fig). Analysis of both IL-1β and LDH secretion confirmed that shRNA knockdown of human CARD8 does not abrogate either IL-1β or LDH release in both infected and VbP-stimulated cells. These findings suggest that functionally active CARD8 is dispensable for NLRP1-mediated HIV-1 icRNA sensing and inflammasome activation. Collectively, these data suggest that NLRP1 expression in THP-1/PMA macrophages leads to cytoplasmic HIV-1 icRNA sensing and IL-1β secretion independent of MAVS, STING, or CARD8 activation.

## NLRP1 detects cytosolic HIV-1 icRNA expression

It was recently reported that NLRP1 binds viral dsRNAs generated during positive-strand RNA virus infection [93]. Positive-strand RNA viruses generate dsRNA replication intermediates in the cytoplasm of infected cells with regions of extensive base pairing [94–96]. Although dsRNA is not thought to be generated in HIV-1-infected cells, HIV-1 unspliced or icRNAs contain extensive secondary structures with RNA duplex regions limited to select sites in the viral genome, such as the rev response element, the 5′ untranslated region including the *trans*-activation response element and dimerization domain, and a hairpin structure in *gag-pol* orf that mediates a frameshift during synthesis of the Gag-Pol polyprotein [97,98]. We sought to investigate if HIV-1 infection of THP-1/PMA macrophages results in the expression of dsRNAs. To this end, we infected THP-1/PMA macrophages with LaiΔenvGFP/G (WT) or Rev-deficient (M10) (MOI = 1) and processed for immunofluorescence analysis for dsRNA expression at 3 dpi with anti-dsRNA specific antibody J2, previously shown by us and others to detect dsRNAs, of at least 40 base pairs in size [99,100]. Immunofluorescence analysis of GFP and dsRNA expression (Fig 7A) and quantification of mean fluorescent intensity per cell (Fig 7B) showed that, while infection efficiency based on GFP expression between WT and M10/HIV-1 was similar (left panel), dsRNA puncta (highlighted by white arrowheads) are only expressed in HIV-1 WT but not M10 (right panel)-infected cells or in EFV-treated HIV-1-infected cells. Cytoplasmic dsRNA puncta was also detected in Lai/YU2*env*-infected MDMs and iMGs (S6A and S6B Fig). Even though direct co-localization of dsRNA and NLRP1 was not observed, distinct cytoplasmic expression of dsRNA suggested that positive dsRNA staining presumably corresponds to intron-containing viral usRNAs instead of msRNAs.

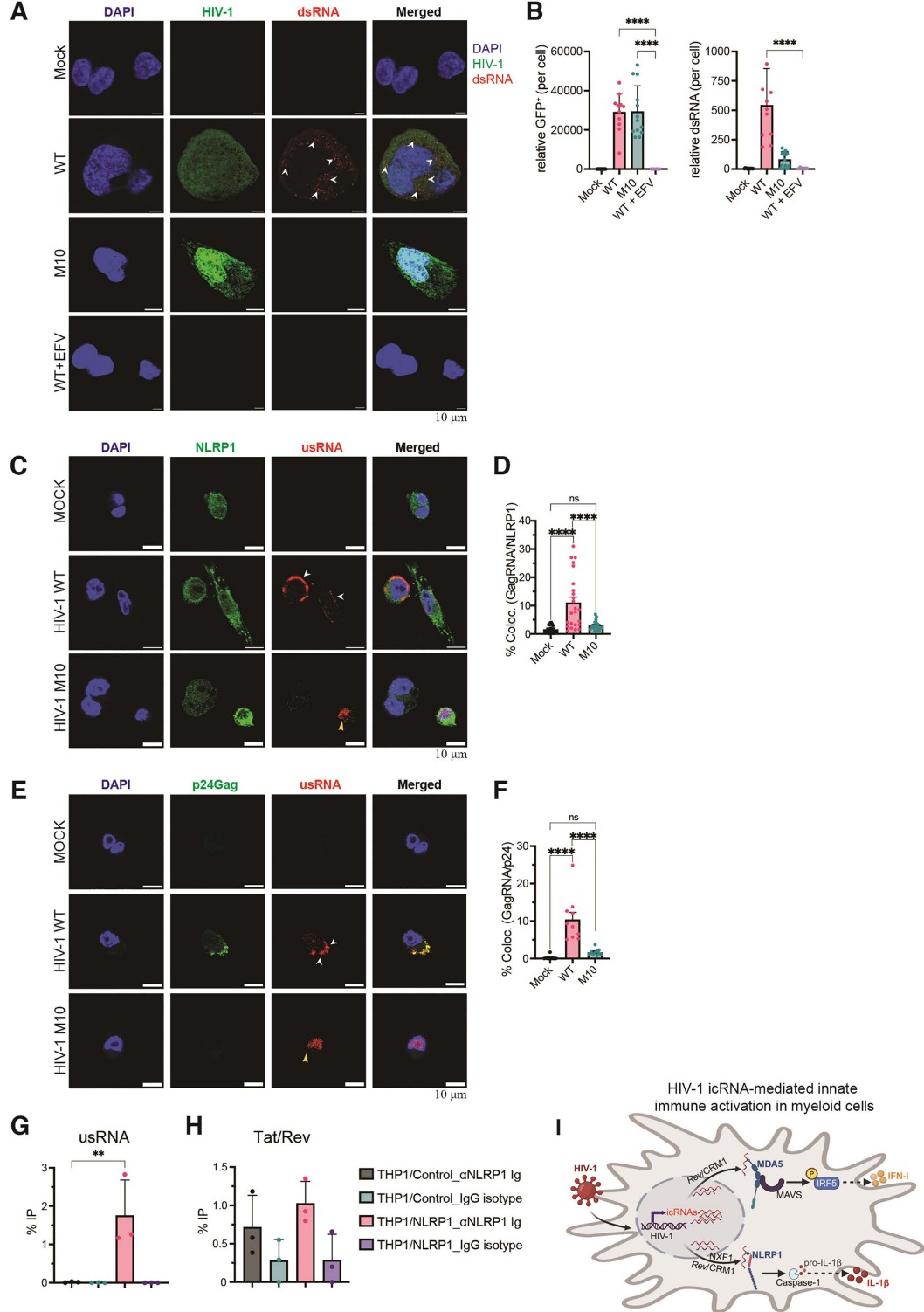

**Fig 7. Cytoplasmic expression of HIV-1 icRNA in macrophages is detected by NLRP1. (A)** Immunofluorescence analysis of PMA-differentiated THP-1 parental (control) macrophages infected with LaiΔenvGFP/G (WT, M10, or WT in the presence of RT inhibitor (EFV), MOI 1), and stained with anti-dsRNA (J2) antibody. The establishment of productive HIV-1 infection in THP1/PMA macrophages was visualized by GFP expression, and nucleus is stained with DAPI. The white arrowheads indicate positive staining of dsRNA puncta. Scale bar represents 5 μm. **(B)** Quantification of relative mean

fluorescent intensity of GFP⁺ cells and dsRNA⁺ puncta. The means ± SEM are shown, and each dot corresponds to an individual image representing a total of 60−80 cells per image per infection condition. **(C–F)** THP-1/NLRP1-PMA macrophages infected with WT or Rev-deficient (M10) LaiΔenvGFP/G, MOI = 1) were stained with anti-NLRP1 antibody **(C)** or anti-p24$^{Gag}$ antibody **(E)** followed by hybridization with Quasar 670-smFISH probes targeting HIV-1 usRNA at day 3 pi. Data are representative of 2 independent experiments. Bar = 10 μm. Quantification of HIV-1 usRNA colocalization with NLRP1 **(D)** or HIV-1 p24$^{Gag}$ **(F)** determined from at least 20 independent fields (representative fields shown in E and G, respectively). Approximately 50−100 infected cells were analyzed. The means ± SEM are shown, and each symbol represents an independent field/cell. **(G, H)** THP-1/parental and THP-1/NLRP1 monocytes infected with LaiΔenvGFP/G (MOI 2) were harvested at 3 dpi, followed by fractionation into cytosolic and nuclear fractions. RT-qPCR quantification of HIV-1 usRNA **(G)**, and tat/rev (msRNA) **(H)** in cytosolic fractions immunoprecipitated with anti-NLRP1 antibody or IgG control. Immunoprecipitated viral RNA copy numbers were determined using standard curves for both usRNA and msRNA, and the percent IP (immunoprecipitated usRNA or msRNA transcripts detected from each IP condition using anti-NLRP1 antibody or control IgG to the input amount in each cell type) is reported as mean ± SEM, where each symbol represents an independent experiment. **(I)** Schematic of the diversity of cytosolic HIV-1 icRNA sensing and innate immune activation pathways in myeloid cells. Image was created in BioRender. Jalloh, S. (2025) https://BioRender.com/q0g4tr1. Statistical significance was determined by one-way ANOVA followed by Dunnett's post-test comparing EFV-treated HIV-WT infection to untreated WT or M10 **(B)**, or IP from anti-NLRP1 antibody in NLRP1-overexpressing THP-1 to control THP-1 cells **(G, H)**. Alternatively, statistical significance was determined by one-way ANOVA followed by Tukey's multiple comparisons test for uninfected (mock), HIV-1/WT, and HIV-1/M10 infected THP1/NLRP1-PMA macrophages **(D, F)**. *P*-values: **** < 0.0001; ** < 0.01; no symbol: not significant (*p* ≥ 0.05). The data underlying this figure can be found in S7 Data.

To determine if NLRP1 co-localizes with HIV-1 usRNA, we utilized single-molecule RNA FISH (smFISH) to specifically label HIV-1 usRNAs using Quasar670-conjugated oligonucleotides (S1 Table) in virus-infected THP-1/NLRP1-PMA macrophages, in tandem with immunofluorescence microscopy to detect endogenous NLRP1 expression (Fig 7C), and HIV-1 p24$^{Gag}$ (Fig 7E). While HIV-1 usRNA was peripherally localized near the plasma membrane in wild-type HIV-1-infected cells (Fig 7C and 7E), HIV-1 usRNA localization was restricted to the nucleus in HIV-1/M10-infected macrophages (Fig 7F). Interestingly, co-localization of NLRP1 and p24$^{Gag}$ was observed with HIV-1 usRNAs in wild-type virus-infected cells, but not in cells infected with Rev-deficient (M10) HIV-1 (Fig 7C and 7E), suggesting that de-novo transcribed cytosolic HIV-1 icRNAs (and not the incoming genome) are co-localized with NLRP1 in productively infected cells. To determine whether NLRP1 can bind to HIV-1 icRNAs, THP-1/control, and THP-1/NLRP1 monocytes were infected with LaiΔenvGFP/G (MOI = 2), followed by fractionation of cellular lysates to nuclear and cytoplasmic fractions (Fig 7G and 7H). RNPs in cytoplasmic fractions were incubated with control IgG or anti-NLRP1 antibody, and co-immunoprecipitation of usRNA (Fig 7G) or msRNA (*tat/rev*) (Fig 7H) was quantified by RT-qPCR. Analysis of cytoplasmic HIV-1 usRNA or msRNA co-immunoprecipitated by anti-NLRP1 antibody compared to control IgG revealed preferential enrichment of HIV-1 usRNA in THP1/NLRP1 cells compared to the parental THP1 cells (Fig 7G). In contrast, no significant difference in immunoprecipitation of HIV-1 msRNA by NLRP1 antibody was observed between NLRP1-overexpressing cells compared to parental cells (Fig 7H). These findings suggest that NLRP1 interacts with HIV-1 usRNA, resulting in inflammasome activation and IL-1β secretion in myeloid cells.

## Discussion

We recently reported that de novo HIV-1 icRNA expression triggers type I IFN responses in macrophages and microglia [34,55,56]. Here, we provide evidence of HIV-1 icRNA-mediated inflammasome activation in myeloid cells. In contrast to MDA5-dependent MAVS-mediated induction of type I IFN responses, which was exclusively dependent on Rev-CRM1-mediated nuclear export of HIV-1 icRNA [57,59,71,82], cytoplasmic expression of HIV-1 icRNA facilitated by either Rev/CRM1 or CTE resulted in NLRP1 inflammasome activation in macrophages (Fig 7I). These findings suggest that the requirements for activation of the disparate inflammatory pathways in HIV-infected macrophages might be governed by distinct cytoplasmic pools of HIV-1 icRNAs. Future studies will address the requirements of distinct sub-cellular localization and/or epitranscriptomic modifications for sensing by MDA5 and NLRP1, respectively.

Though NLRP1 was the first identified inflammasome [101], mechanisms of pathogen-induced NLRP1 inflammasome activation are yet to be clearly defined because of the uncertainty in pathogen ligands that engage NLRP1 [102]. NLRP1 inflammasome activation has been attributed to distinct host and extracellular stimuli, including ribotoxic and reductive

stress [90,103], viral protease activity [104,105], and viral dsRNA synthesis [93,106]. Here, we report that HIV icRNA in the cytosol is another trigger of NLRP1 inflammasome activation. NLRP1 contains a caspase-1 activation and recruitment domain (CARD), a pyrin domain, a leucine-rich repeat (LRR), a conserved in UNC5, PIDD, and ankyrins (UPA) subdomain, and a function-to-find-domain (FIIND) [107]. NLRP1 undergoes autoproteolytic processing in its FIIND domain to yield two non-covalently linked protein fragments (large N-terminal, NLRP1NT and the small C-terminal fragment, NLRP1CT) [107]. The NLRP1CT fragment, which contains the inflammasome-forming CARD, is maintained in an auto-inhibitory state under homeostatic conditions until N-terminal perturbation and proteasomal degradation of the NLRP1NT fragment releases the bioactive C-terminal UPA-CARD fragment to oligomerize and form a platform for caspase-1 recruitment and activation [107]. Binding of PAMPs such as viral dsRNA to LRR of NLRP1 is hypothesized to induce ATPase-driven conformational change that releases NLRP1CT and results in inflammasome activation [93]. While dsRNA intermediates are generated during the replication cycle of diverse RNA and DNA viruses [94,96], to our knowledge, these cytoplasmic RNA puncta recognized by an anti-dsRNA antibody have not been previously reported in HIV-infected macrophages. While HIV-1 icRNA co-localized with NLRP1 in the cytoplasm, the exact mechanism by which cytosolic expression of HIV-1 icRNA activates NLRP1 and whether specific HIV-1 icRNA features, such as the extent of duplex RNA regions, RNA modifications, or unique cytoplasmic localization and whether additional co-factors are required remain to be fully elucidated.

In myeloid cells, NLRP3, NLRC4, and CARD8 have all been shown to activate inflammasomes and stimulate IL-1β release upon sensing cytoplasmic HIV-1 proteins or RNA [41,42,44,45,47,50,52–54,108]. While previous studies using in vitro transcribed single-stranded RNA containing GU-rich sequences from HIV-1 LTR have suggested that endocytosed HIV-1 gRNA (TLR 7/8 ligand) mediates IL-1β secretion and pyroptotic cell death in glial cells [44], our findings suggest that de novo cytoplasmic expression of HIV-1 icRNA is required for NLRP1 inflammasome activation, since pretreatment of cells with RT or integrase inhibitors abrogated IL-1β secretion. Previous studies have also linked HIV-1-infection-induced inflammasome activation to CARD8 expression in macrophages [52–54,109] and demonstrated that pyroptotic cell death and IL-1β secretion either required NNRTI-induced dimerization of Gag-Pol or Gag-Pol overexpresion and subsequent PR activation [91,92]. In contrast, our findings suggest that Gag-Pol deletion or inhibition of PR activity (Fig 2) only partially inhibited HIV-1-infection-induced NLRP1-mediated IL-1β secretion (Fig 2). Notably, the reduced levels of IL-1β secretion from HIV-1/ΔGag-Pol-infected MDMs compared to HIV-1/WT-infected MDMs suggest that both CARD8 and NLRP1 inflammasome activation contribute to IL-1β secretion in HIV-1-infected cells. Importantly, these findings suggest late steps in the HIV-1 life cycle and virus assembly are subjected to multiple pathogen sensing pathways, persistent activation of which might contribute to HIV-1-induced inflammatory pathologies.

Inflammasome activation is correlated with HIV-induced neurocognitive dysfunction [28,36,48], and HIV infection of CNS-resident macrophages and microglia has been shown to drive sustained proinflammatory cytokine induction such as IL-1β and TNFα, and amyloid-β deposition leading to neuronal injury and brain dysfunction [110–113]. Thus, regulating inflammasome activation might be essential to prevent protracted inflammation and chronic disorders such as HAND [25,114–116]. Understanding the distinct mechanisms by which inflammasomes are activated and identification of specific targets might lead to the development of therapeutic strategies to dampen the chronic inflammation and immune dysregulation observed in PWH. Previous efforts have predominantly focused on targeting steps in the virus life cycle [117,118] to suppress HIV-1-induced chronic inflammation. However, emerging interventions targeting host drivers of inflammation provide an exciting avenue for future therapeutics [119–121] against HIV-associated non-AIDS complications.

## Materials and methods

### Ethics statement

This research has been determined to be exempt by the Institutional Review Board of the Boston University Medical Center since it does not meet the definition of human subjects research, as all human samples were collected in an anonymous fashion and no identifiable private information was collected.

## Viruses

HIV-1 replication-competent molecular clone, Lai/YU-2env, single-round GFP expressing reporter virus constructs, LaiΔenvGFP (GFP in place of the *nef* orf), and *rev*-deficient mutant LaiΔenvGFP M10, have been described previously [56]. LaiΔenvGFPΔGag-pol proviral plasmid, includes an additional deletion between N-terminus of CA and IN, and has been described previously [56]. The LaiΔenvGFP M10-4xCTE construct was generated by PCR-amplifying CTE sequence from the plasmid, pCR4-CTE (Addgene #36868) using the following primers (forward: TTTTTGGATCCACTATAGGG-CGAATTGAATTTAGCG; reverse: GGATAACAAT TTCACACAGG AAACAGCTAT GAC) and inserted into the LaiΔen-vGFP M10 proviral plasmid using BclI and NheI restriction enzymes. Replication-competent viruses were derived from HEK293T cells via calcium phosphate-mediated transient transfection of Lai/YU-2env. Single-round viruses pseudotyped with VSV-G were generated from HEK293T cells via co-transfection of LaiΔenvGFP proviral plasmids and VSV-G expression plasmid (pHCMV-G). For generation of LaiΔenvGFP/M10, LaiΔenvGFP/M10-4XCTE or LaiΔenvGFPΔGag-Pol, the HIV-1 Gag-Pol packaging plasmid (psPAX2) was co-transfected with the *rev* or *gag*-deficient proviral plasmids. $SIV_{mac}$ Vpx containing VLPs were generated from HEK293T cells via co-transfection of SIV3+, and VSV-G expression plasmid, as described previously [56]. NLRP1-expressing lentivector (LV-NLRP1) was derived by recombining the pDONR-NLRP1 plasmid (Addgene #166823, [89]) with the pLenti CMV puro DEST w118-1 plasmid (Addgene #17452, [122]) via LR clonase (Gateway LR Clonase II Enzyme mix, Invitrogen # 11791020). Lentiviruses expressing NLRP1 (or vector control) were produced via co-transfection of HEK293T cells with NLRP1 expressing lentivector, psPAX2 and pHCMV-G. Virus-containing cell culture supernatants were harvested 2 days post-transfection, passed through 0.45 µm filters, and purified and concentrated by ultracentrifugation on a 20% sucrose cushion (24,000 rpm and 4°C for 2 h with a SW32Ti or SW28 rotor (Beckman Coulter)). Virus pellets were resuspended in PBS, aliquoted, and stored at −80°C until use. The capsid content of HIV-1 was determined by a $p24^{gag}$ ELISA and virus titer was measured on TZM-bl by measuring β-gal activity as previously described [56].

## Cells and viruses

HEK293T (ATCC) were maintained in DMEM (Gibco) containing 10% heat-inactivated FBS (Gibco) and 1% pen/strep (Gibco). THP-1 and TZMbl cell lines (NIH AIDS reagent program) were maintained in DMEM (Gibco) containing 10% FBS and 1% pen/strep. All cell lines are regularly tested for mycoplasma contamination and confirmed negative. Where indicated, THP-1 cells were stimulated with PMA (100 nM, Sigma-Aldrich) for 48 hours and treated with 10 ng/mL TNFα (Peprotech) for the duration of infection. To generate shRNA or NLRP1 expressing THP-1 cells, cells were transduced with lentivectors via spinoculation in the presence of polybrene (10 µg/ml, Sigma-Aldrich) before selection and mainte-nance with 2 µg/mL puromycin (Invivogen), plus 400 µg/mL hygromycin (Gibco) where indicated.

## Isolation and differentiation of primary human macrophages

Peripheral blood mononuclear cells were isolated from de-identified leukopacks obtained from NY Biologics as described previously [123]. Monocytes were positively isolated using antibody-coated beads against CD14 (Miltenyi, Cat#130-050-201). Monocytes were cultured in 20 ng/mL M-CSF (Peprotech, catalog #300-25B), heat-inactivated 10% human AB serum (Sigma-Aldrich), 1% pen/strep in RPMI1640 (Gibco) for 5–6 days for differentiation into macrophages as previously described [56].

## Generation of human iPSC-derived microglia

Two iPSC lines, BU1 and BU3 (herein referred to as iMG #1 and iMG #2, respectively) were first differentiated toward a mesodermal, hematopoietic lineage on a matrigel-coated 6-well plate in mTeSR (Stem Cell Technologies). Next, dif-ferentiation was initiated by culturing cells in a cocktail of growth factors, as described in [124], maintained in a hypoxic

incubator (5% $O_2$). Cells were continually fed every 2 days and maintained in hypoxia up to day 6, varying the components and concentrations of the differentiation cocktail [124]. On day 6, the media composition was changed for the next 6 days (day 6–12), with the addition of cytokines, maintaining in hypoxia condition. On day 8, the cells are fed and returned to normoxia for the rest of the differentiation. At day 12, hematopoietic cells are harvested from the culture supernatant. This population typically contains 25%–65% (average 43%) CD34+CD45+ progenitor cells. Subsequently, the hematopoietic cells harvested are differentiated toward microglia lineage using STEMdiff Microglia Differentiation Kit and STEMdiff Microglia Maturation Kit.

### Infections

Cells were spinoculated with HIV-1 (1 h at RT and 1,100$g$) at the indicated multiplicity of infection, cultured for 2−3 h at 37°C, washed to remove unbound virus particles, and cultured in fresh media for 2−3 days. Cells were seeded at a density of 0.5 million MDMs or iMGs/well or 0.25 million THP1/PMA macrophages in 12-well plates. For indicated experiments, cells were pre-treated for 30 min with inhibitors at 37°C before infection and maintained throughout the culture. Inhibitors used in this study are a reverse transcription inhibitor efavirenz (1 μM, NIH AIDS Research and Reference Reagent Program, catalog #4624), an integrase inhibitor raltegravir (30 μM or 60 μM, Selleck Chemicals, catalog #50-615-1), protease inhibitors; saquinavir, darunavir, and lopinavir (1 μM, NIH AIDS Research and Reference Reagent Program), spironolactone (30 μM; Selleck Chemicals), or KPT-335 (0.1 μM, verdinexor; Selleck Chemicals), a pan-caspase inhibitor zVAD-fmk (30 μM, Sigma, catalog #219007), a caspase-1 inhibitor VX-765 (1 μM, InvivoGen, catalog #inh-vx765-1), ValboroPro NLRP1/CARD8 inducer VbP (10 or 100 μM, InvivoGen, #tlrl-vbp-10). Where indicated, TNFα was added (10 ng/ml) as a priming signal in THP-1/PMA cells immediately after washing unbound virus or at 24 h before harvest. At the indicated time points, culture supernatants were harvested for ELISA, and cells were collected for RNA or protein analysis.

### siRNA transfection and functional knockdown

MDMs were seeded overnight at a density of 0.5 or 0.25 million cells per well in 12- or 24-well plates, respectively. Scramble control siRNAs or specific siRNAs against indicated host targets were transfected at 50 nM for 2 days using the TransIT-X2 dynamic delivery system (Mirus, #MIR6000). At day 2 post-transfection, cells were either infected with HIV-1 (as described above), or stimulated as follows to determine functional knockdown: AIM2 (cells were treated with ultra-pure LPS (100 ng/m, Invivogen) for 2 h, followed by transfection with linearized DNA (1 μg/mL) for 4 h); NLRP1 and CARD8 (cells were primed with Pam3CSK4 (0.5 μg/mL, InvivoGen) for 4 h, followed by stimulation with Val-boroPro (10 μM, InvivoGen) for 24 h; NLRP3 and caspase-1 (cells were primed with ATP (5 mM, Thermo Scientific) for 6 h, followed by activation with nigericin (10 μM) for 60 min; Caspase-4 (cells were transfected with ultra-pure LPS (5 μg/ml, InvivoGen) for 6 hrs; UNC93B1 (cells were treated with Resiquimod (5 μg/ml, Invivogen) for 24 h. At the indicated timepoints, culture supernatants were harvested for IL-1β and IP-10 ELISA, and cells were lysed for RNA or protein analysis.

### Quantitative RT-PCR and viral RNA analysis

At 48−72 h post-infection, cells were harvested for RNA isolation using RNeasy RNA isolation kits (QIAGEN, catalog # 74104) and cDNA was synthesized using oligo dT primers and Superscript III RT (Invitrogen, catalog #18080-051). cDNA corresponding to 100 ng of RNA was analyzed by qRT-PCR using SYBR green (Thermo Scientific, catalog # FERK0241) to quantify host mRNA levels and HIV transcripts using the primer sets listed in Table 1. The data were normalized to input (for immunoprecipitation) or GAPDH control (as infection control). The $C_T$ value was normalized to that of GAPDH and represented as a relative value to mock or drug control, where indicated, using the $2^{-\Delta\Delta C_T}$ method as previously described [125,126].

## ELISA

Secreted IL-1β or IP-10 in culture supernatants was measured using a commercially available Human IL-1 beta/IL-1F2 DuoSet ELISA kit (R&D Systems, catalog #DY201) or Human IP-10 ELISA Set (BD Biosciences), respectively, according to directions provided by the manufacturer. HIV-1 p24$^{Gag}$ levels in the culture supernatants were assessed using an in-house p24$^{Gag}$ ELISA [56].

## Flow cytometry

Cells were analyzed using BD LSRII or Cytek Aurora instruments with the help of Boston University Flow Cytometry Core who provided instrumentation and technical support. Data analysis was performed using FlowJo software (FlowJo). For cell surface antigen staining, cells were pre-chilled at 4°C for 30 min prior to staining. Antibody solution was added at indicated dilution in 2% NCS (normal calf serum, Invitrogen, catalog #26170043) in PBS (catalog #14190-250) for 30 min at 4°C. Cells were washed once with 2% NCS/PBS and fixed in 4% paraformaldehyde (PFA) in PBS (Boston Bioproducts) for at least 30 min at 4°C. For intracellular antigen staining, cells were fixed with 4% PFA, and permeabilized using Perm/Wash Buffer (BD Biosciences, catalog #554723) for at least 15 min at RT. Cells were stained at the indicated dilution of antibody in Perm/Wash buffer for 30 min at 4°C. Cells were washed once with Perm/Wash and resuspended in 2% NCS/PBS.

## Antibodies

To characterize iMGs following differentiation, the following antibodies were used; mouse anti-CD11b (BioLegend; #301410; 1:20), mouse anti-CD45 (BD Biosciences; #555483; 1:10), mouse anti-P2RY12 (Biolegend, #392105, 1:50), mouse anti-TREM2 (Biolegend, #824803, 1:50). Intracellular p24$^{Gag}$ was stained using FITC-conjugated mouse anti-p24$^{Gag}$ monoclonal antibody (KC57; Beckman Coulter, catalog # 6604665, 1:25) as previously described [126]. For all experiments, cell viability was assessed using Zombie NIR Fixable Viability Kit (Biolegend, #423106, 1:250).

## Imaging

For MDMs and THP-1, cells were cultured and infected on coverslips (LabTekII) and were washed and fixed with 4% PFA. Cells were then permeabilized with 0.1% TritonX100 in PBS, and stained as follows. Permeabilized cells were incubated

**Table 1. qRT-PCR primers.**

| Gene | Forward | Reverse |
|------|---------|---------|
| gRNA | TGTGTGCCCGTCTGTTGTGT | CTCTCCTTCTAGCCTCCGCT |
| Tat/Rev | GCGACGAAGACCTCCTCAG | GAGGTGGGTTGCTTTGATAGAGA |
| GAPDH | CAAGATCATCAGCAATGCCT | AGGGATGATGTTCTGGAGAG |
| Caspase-1 | GCTGAGGTTGACATCACAGGCA | TGCTGTCAGAGGTCTTGTGCTC |
| Caspase-4 | TTGCTTTCTGCTCTTCAACG | GTGTGATGAAGATAGAGCCCATT |
| AIM2 | GCTGCACCAAAAGTCTCTCCTC | CTGCTTGCCTTCTTGGGTCTCA |
| NLRP1 | ATTGAGGGCAGGCAGCACAGAT | CTCCTTCAGGTTTCTGGTGACC |
| NLRP3 | CCACAAGATCGTGAGAAAACCC | CGGTCCTATGTGCTCGTCA |
| IL-1β | AAACAGATGAAGTGCTCCTTCC | AAGATGAAGGGAAAGAAGGTGC |
| IL-18 | GACCAAGGAAATCGGCCTCTA | ACCTCTAGGCTGGCTATCTTTATACATAC |
| ZBP1 | TGGTCATCGCCCAAGCACTG | GGCGGTAAATCGTCCATGCT |
| MAVS | GTACCCGAGTCTCGTTTC | GCAGAATCTCTACAACATCC |
| STING | ACTGTGGGGTGCCTGATAAC | TGGCAAACAAAGTCTGCAAG |
| UNC93B1 | TGATCCTGCACTACGACGAG | GCGAGGAACATCATCCACTT |
| RIG-I | ATCCCAGTGTATGAACAGCAG | GCCTGTAACTCTATACCCATGTC |
| CARD8 | CTGAAGGAAATGTGGATGTTGAGT | CCACAGATACCAGCCAGCAGT |

overnight at 4°C with rabbit polyclonal antibody directed against C-terminal NLRP1 (Sigma, #HPA064431) at 1:500 dilution, followed by secondary antibody staining using goat anti-rabbit-AF568 (Invitrogen, #A21069, 1:200) for 1 h at RT. For dsRNA staining [127], a mouse anti-dsRNA (clone J2, Cell Signaling, #76651) antibody was used at 1:500 dilution overnight, followed by goat anti-mouse-AF647 (Invitrogen, #21237, 1:200) for visualization. For nuclear staining, 4′,6-diamidino-2-phenylindole (DAPI; Sigma-Aldrich) was used at 200 ng/ml for 15 min at RT. After each step, cells were washed three times in PBS. Images were acquired using a Leica SP5 confocal microscope and analyzed using ImageJ software.

## Immunoblot analysis

To assess expression of host and viral proteins, cell lysates containing 15–30 μg total protein were separated by SDS-PAGE, transferred to nitrocellulose membranes, and the membranes were blocked with a 1:1 mix of PBS/Li-Cor Odyssey Blocking Buffer (Li-Cor, catalog # 927-70001) in PBS (Invitrogen, catalog #14190-250) for 1–2 h at RT. Blots were then probed with the following primary antibodies overnight at 4°C; human HIV-Ig (NIH AIDS reagent program, #3957, 1:1000), mouse anti-SAMHD1 (Abcam, # ab67820, 1:1000), rabbit anti-phospho-SAMHD1 (Cell Signaling, #15,038, 1:1000), mouse anti-NLRP1 (Biolegend, #6709802, 1:1000), rabbit anti-caspase-1 (Sigma, #06503-I, 1:1000), rabbit anti-CARD8 (Abcam, #ab194585, 1:1000), rabbit anti-MAVS (Invitrogen, #PA5-17256, 1:1000), rabbit anti-STING (Cell Signaling, #13647, 1:1000). Blots were washed 3 times for 5 min with PBS-T (0.05% Tween-20 in PBS). Specific staining was visualized with secondary antibodies, goat anti-mouse-IgG-DyLight 680 (Invitrogen, #35518, 1:20000), goat anti-rabbit-IgG-DyLight 800 (Invitrogen, #SA5-35571, 1:20000), or goat anti-human DyLight 800 (Rockland, #609-145-123) for 1–3 h at RT. As loading controls, β-actin was probed using mouse anti-β-actin (Invitrogen, #AM4302, 1:5000). Membranes were scanned with an Odessy scanner (Li-Cor).

## IFN bioassay

Levels of bioactive type I IFN secreted from HIV-1-infected macrophages and microglia were measured using a HEK293 ISRE-luc reporter cell line [126,128]. Briefly, HEK293 ISRE-luc cells ($8 \times 10^4$) were incubated with cell culture supernatants for 21 h. Cells were lysed in BrightGlo lysis buffer (Promega, catalog #E2661), and luciferase activity in the cell lysates was analyzed with Bright-Glo Luciferase System (Promega, catalog #E2620). Serial dilutions of recombinant IFN$_{\alpha2}$ ranging from 200 to 0.39 units/ml (PBL Interferon Source, catalog #11100-1) were used for generating a standard curve.

## LDH release assay

Release of LDH in culture medium was quantified using the CytoTox 96 Non-Radioactive Cytotoxicity Assay (Promega, #PR-G1780) according to the manufacturer's instructions. All samples were lysed in 0.5% Triton X-100, and LDH release was normalized to mock (uninfected or unstimulated) controls.

## FLICA assay

PMA-differentiated THP1 cells were either treated with virus-free media (Mock) or infected with LaiΔenvGFP/G WT (MOI = 1) in the absence (DMSO) or presence of RT inhibitor, Efavirenz (EFV, 1 μM) for 3 days. As positive control, cells were transfected with lipopolysaccharide (LPS, 1 μg) for 24 h, followed by treatment with Nigericin (Ngc, 20 μM) for 30 min. Cells were then stained with FLICA Caspase-1 (ImmunoChemistry Technologies, #9,122) for 1 h, following manufacturer's instructions. Cells were then collected, washed three times, fixed with 4% paraformaldehyde, and analyzed by flow cytometry. Percent infection was quantified based on GFP expression, and caspase-1 activation determined by YVAD-660 expression.

## RNA-immunoprecipitation (RNA-IP)

Infected cells were washed twice with PBS, and where indicated, followed by cellular fractionation using PARIS kit (Invitrogen, #AM192), according to the manufacturer's guide. Whole cell lysates or cytoplasmic fractions were

immunoprecipitated following a recently described protocol [129]. Briefly, protein A Dynabeads (Invitrogen) were conjugated with human anti-NLRP1 antibody (Novus Biologicals, #NB100-56148SS, diluted at 10 µg per 50 µL beads) for 1 hour while rotating at RT. Unfractionated cells were lysed in NET-2 buffer (50 mM Tris-HCl, 150 mM NaCl, 0.05% NP-40), and fractionated samples were lysed using cellular fractionation buffer supplied in PARIS kit. After bead preparation, NLRP1-conjugated beads were incubated with cell lysates at 4°C for 2 h while rotating. This is followed by magnetic bead separation, and washing thrice in NET-2 buffer. After RNA extraction using the TRIzol reagent (Invitrogen), RNA samples were treated with DNase using Turbo DNA-free kit (Invitrogen, #AM1907), and cDNA synthesis, followed by qRT-PCR was carried out as described above.

### smFISH

**Probe designs.** smFISH probes used to detect HIV-1 Lai usRNA consisted of a set of 48 oligonucleotides, each with a length of 20 nt and labeled with Quasar 670 (see S1 Table). Probes were designed in-house, with the 3′-end of each probe modified with an amine group and coupled to Quasar 670 (Stellaris).

**Hybridization.** PMA-differentiated THP-1/NLRP1 macrophages were cultured on glass coverslips, infected at indicated MOIs for 2−3 days, followed by fixation with 3.7% formaldehyde (10 min, RT) and permeabilization with 0.1% nuclease-free Tween-20 (15 min, RT). After permeabilization, coverslips were blocked with 1% RNase-free BSA (15 min, RT) and incubated with either mouse rabbit anti-NLRP1 (Novus Biologicals, #NB100-56147, 1:500) or rabbit anti-p24$^{Gag}$ (Immuno Diagnostics, #1303, 1:500) in 1×PBS containing 0.1% BSA for 1 h at room temperature. Coverslips were then washed three times with 1×PBS, followed by incubation with goat anti-rabbit AlexaFluor568 (Invitrogen, #2935303, 1:200) in 1×PBS containing 0.1% BSA for 1 h at RT. Each coverslip was then washed three times with 1×PBS, followed by fixation with 3.7% formaldehyde for 10 min at RT. Coverslips were rinsed with 1×PBS, equilibrated in Wash Buffer A (5 min, RT), and then placed faced down onto a 100 µl droplet of hybridization buffer containing the Quasar 670 conjugated HIV-1 Gag RNA probes (125 nM working concentration), and incubated at 37°C overnight in a humid chamber. Following hybridization, the coverslips were washed once with Wash Buffer A for 30 min at 37°C, followed by DAPI nuclear staining (5 ng/ml) for 30 min at 37°C. The coverslips were then equilibrated with Wash Buffer B for 5 mins at RT, followed by mounting using Vectashield Mounting Medium.

**Image acquisition and analysis.** Images were acquired using Leica SP5 Confocal Microscope (63x oil immersion objective; numerical aperture 1.4). Stacks of images of 20 layers with 0.25 µm interval at 100–2,000 ms exposure for each channel were acquired. At least two representative coverslips per infection condition were examined, and approximately 20 regions/fields of interest were acquired. For cell fluorescence intensity measurements, region of interest was drawn manually around each cell using DIC and DAPI channels, and co-localization between the RNA-specific channel (Quasar 670) and the antibody-specific channel (AlexaFluor568) was quantified using the JACoP plugin in ImageJ (Fiji).

### Statistics

All the statistical analysis was performed using GraphPad Prism 9. *P*-values were calculated using one-way ANOVA followed by the Tukey–Kramer post-test or the Dunnett's post-test. Symbols represent, *: $p < 0.05$, **: $p < 0.01$, ***: $p < 0.001$, ****: $p < 0.0001$. No symbol: not significant ($p \geq 0.05$).

### Supporting information

**S1 Fig. HIV-1 icRNA-dependent and protease-independent inflammasome activation in HIV-1-infected-macrophages.** Quantitation of p24$^{Gag}$ release (**A** and **C**) and IL-1β secretion (**B** and **D**) by ELISA in culture supernatants from MDMs that were either stimulated with or without TNFα (10 ng/ml), pre-treated with either Vpx-VLPs or deoxynucleosides (dNs) before infection with LaiΔenvGFP/G (WT, ΔGag-pol, or M10), was quantified at d3pi. The means ± SEM of 3

independent experiments are shown with cells derived from 3 independent donors. Statistical significance was determined by one-way ANOVA followed by Tukey's multiple comparisons test (**A** and **B**), or the Kruskal–Wallis test, comparing uninfected (mock) to WT or mutant HIV-infected MDMs in both Vpx and dNs-treated conditions. *P*-values: \*\*\*< 0.001; \*\*< 0.01; \*< 0.05; no symbol: not significant ($p \geq 0.05$). The data underlying this figure can be found in S1 Supplementary Data. (PDF)

**S2 Fig. HIV-1 infection of iMGs induces IL-1β secretion.** (**A**) Immunoblotting (left panel) and quantitation (right panel) for unphosphorylated and phosphorylated forms of SAMHD1 in iMGs from 2 iPSC lines (BU1 and BU3) and MDMs from two different donors ± Vpx-VLPs. (**B**) Representative immunofluorescence images of iMGs infected with replication-competent Lai/YU-2env (MOI = 1) and analyzed by confocal microscopy. Cells were stained for intracellular p24[Gag], IBA1, and DAPI. Scale bar represents 10 μm. (**C**) Infection of iMGs (BU3) with LaiΔenvGFP/G (MOI 1) in the absence (no treatment, NT) or presence of EFV (1μM) for 3 days, analyzed by flow cytometry for GFP expression. (**D**) ELISA for p24[Gag] (left panel) and IL-1β (right panel) expression in supernatants from LaiΔenvGFP/G-infected iMGs. The data is reported as mean ± SEM from 3 independent experiments from iMGs derived from 2 donor lines. Statistical significance was determined by one-way ANOVA followed by Dunnett's post-test compared to HIV-infected iMGs (**D**). *P*-values: \*\*\*\*< 0.0001; no symbol: not significant ($p \geq 0.05$). The data underlying this figure can be found in S2 Supplementary Data. (PDF)

**S3 Fig. Functional knock-down of inflammasome components in macrophages.** (**A** and **B**) Transient knockdown of AIM2, NLRP1, NLRP3, CARD8, Caspase-1, Caspase-4, and UNC93B1 by siRNA transfection in primary MDMs from 3 donors, followed by stimulation with respective agonists for each target (described below). Culture media was harvested at the time points post-stimulation described below. Quantification of IL-1β released in the culture supernatants determined by ELISA (**A**). ISG induction post-stimulation was determined by quantifying secreted IP-10 by ELISA (**B**). Stimulations post-siRNA knockdown of MDMs: AIM2, cells were treated with ultra-pure LPS (100 ng/mL) for 2 h, followed by transfection with linearized DNA (1 μg/mL) for 4 h; NLRP1 and CARD8, cells were primed with Pam3CSK4 (0.5 μg/mL) for 4 h, followed by stimulation with VbP (10 μM) for 24 h; NLRP3 and Caspase-1, cells were primed with ATP (5 mM) for 6 h, followed by activation with nigericin (10 μM) for 60 min; Caspase-4, cells were transfected with ultra-pure LPS (5 μg/ml) for 6 h; UNC93B1, cells were treated with Resiquimod (5 μg/ml) for 24 h. For each knockdown and stimulation, IL-1β or IP-10 secretion from siControl-transfected MDMs was set at 100, and the fold decrease upon inflammasome component knockdown was reported. The means ± SEM are shown, and each symbol represents an independent donor. The data underlying this figure can be found in S3 Supplementary Data. (PDF)

**S4 Fig. Cytoplasmic expression of NLRP1 in primary human macrophages and iPSC-microglia.** Representative immunofluorescence images of NLRP1 expression in THP-1/PMA macrophages, THP1/PMA/NLRP1 macrophages, MDMs, and iMGs. Scale bar represents 5 μm. (PDF)

**S5 Fig. HIV-1 induces IL-1β secretion in a CARD8-independent manner in NLRP1-expressing THP1 macrophages.** (**A**, **B**) PMA-differentiated THP-1/NLRP1 cells were either treated with virus-free media (Mock), infected with LaiΔenvGFP/G WT (MOI = 1) in the absence (DMSO) or presence of RT inhibitor, Efavirenz (EFV) for 3 days, or transfected with lipopolysaccharide (LPS, 1 μg) for 24 h, followed by treatment with Nigericin (Ngc, 20 μM) for 30 min. (**A**) Representative flow cytometry profiles of THP1/NLRP1 cells showing expression of GFP (x-axis) and caspase-1 activation determined by YVAD-660 expression (y-axis). (**B**) Quantification of % GFP expression (left panel), and % caspase-1 activation (right panel). (**C**–**E**) PMA-differentiated THP-1/parental macrophages were treated with 10 ng/mL TNFα and co-infected with wildtype (WT) or protease-inactive mutant (ΔPR) single-cycle HIV-1 and SIV3+/Vpx VLPs for 48h before medium change

and treatment with 5 µM RPV or DMSO (control). Graphs show (**C**) GFP expression, (**D**) IL-1β secretion, and (**E**) LDH release after 24h drug treatment. (**F**, **G**) western blot showing CARD8 expression (F) or NLRP1 expression (G) in (1) parental THP-1 cells with knockdown (KD) control (shScramble), (2) parental THP-1 cells with CARD8 KD (shCARD8), (3) THP-1/NLRP1 cells without KD (shScramble), or (4) THP-1/NLRP1 cells with CARD8 KD (shCARD8), with or without differentiation with PMA (48h). (**H–J**) The 4 lines of THP-1/PMA macrophages shown in (F) and (G) were treated with 10 ng/mL TNFα and infected with LaiΔenvGFP/G (MOI = 1) or treated with VbP (100 µM). Graphs show (H) GFP expression, (I) IL-1β secretion, and (J) normalized LDH release 72h post-infection (or 24h post-VbP treatment). Statistical significance was determined by one-way ANOVA followed by Dunnett's post-test compared to DMSO-treated uninfected control (mock) (D, E). *P*-values: ****< 0.0001; no symbol: not significant ($p \geq 0.05$). The data underlying this figure can be found in S4 Supplementary Data.
(PDF)

**S6 Fig. Cytoplasmic expression of dsRNA puncta in HIV-1-infected macrophages and iMGs.** (**A, B**) Representative immunofluorescence images of MDMs (A) or iMGs (B) infected with Lai/YU-2env (MOI = 1), and stained for intracellular p24$^{Gag}$, dsRNA, and DAPI at 3 dpi. The white arrowheads indicate dsRNA puncta. Scale bar represents 20 µm.
(PDF)

**S1 Raw Images. Original western blots of expression of inflammasomes and adaptors in MDMs and THP/PMA macrophages.**
(PDF)

**S1 Table. smFISH Probes for HIV-1 LAI Gag Transcripts.**
(XLSX)

**S1 Data. Data underlying Fig 1.**
(ZIP)

**S2 Data. Data underlying Fig 2.**
(ZIP)

**S3 Data. Data underlying Fig 3.**
(ZIP)

**S4 Data. Data underlying Fig 4.**
(ZIP)

**S5 Data. Data underlying Fig 5.**
(ZIP)

**S6 Data. Data underlying Fig 6.**
(ZIP)

**S7 Data. Data underlying Fig 7.**
(ZIP)

**S1 Supplementary Data. Data underlying S1 Fig.**
(ZIP)

**S2 Supplementary Data. Data underlying S2 Fig.**
(ZIP)

**S3 Supplementary Data. Data underlying S3 Fig.**
(ZIP)

**S4 Supplementary Data. Data underlying S5 Fig.**
(ZIP)

## Acknowledgments

We thank the BUMC Flow Cytometry Core and the Cellular Imaging Core for technical assistance.

## Author contributions

**Conceptualization:** Suryaram Gummuluru.

**Data curation:** Sallieu Jalloh, Ivy K. Hughes, Aldana D. Gojanovich.

**Formal analysis:** Sallieu Jalloh, Ivy K. Hughes.

**Funding acquisition:** Andrew J. Henderson, Gustavo Mostoslavsky, Suryaram Gummuluru.

**Investigation:** Sallieu Jalloh.

**Methodology:** Sallieu Jalloh, Hisashi Akiyama, Aldana D. Gojanovich, Andres A. Quiñones-Molina, Mengwei Yang.

**Project administration:** Suryaram Gummuluru.

**Supervision:** Hisashi Akiyama, Andrew J. Henderson, Gustavo Mostoslavsky, Suryaram Gummuluru.

**Writing – original draft:** Sallieu Jalloh, Ivy K. Hughes.

**Writing – review & editing:** Sallieu Jalloh, Ivy K. Hughes, Andrew J. Henderson, Gustavo Mostoslavsky, Suryaram Gummuluru.

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
