## [Editor Report · Decision Letter 0]

15 Jan 2025

Dear Dr Gummuluru, 

Thank you for submitting your manuscript entitled "Expression of Intron-containing HIV-1 RNA Induces NLRP1 Inflammasome Activation in Myeloid Cells" for consideration as a Research Article by PLOS Biology. I am really sorry for the delay. 

Your manuscript has now been evaluated by the PLOS Biology editorial staff, as well as by an academic editor with relevant expertise, and I am writing to let you know that we would like to send your submission out for external peer review.

Once your full submission is complete, your paper will undergo a series of checks in preparation for peer review. After your manuscript has passed the checks it will be sent out for review. To provide the metadata for your submission, please Login to Editorial Manager (https://www.editorialmanager.com/pbiology) within two working days, i.e. by Jan 17 2025 11:59PM.

Kind regards,

Melissa

Melissa Vazquez Hernandez, Ph.D.

Associate Editor

PLOS Biology

---

## [Decision Letter · Decision Letter 1]

13 Feb 2025

Dear Dr Gummuluru,

Thank you for your patience while your manuscript "Expression of Intron-containing HIV-1 RNA Induces NLRP1 Inflammasome Activation in Myeloid Cells" was peer-reviewed at PLOS Biology. It has now been evaluated by the PLOS Biology editors, an Academic Editor with relevant expertise, and by three independent reviewers. 

In light of the reviews, which you will find at the end of this email, we would like to invite you to revise the work to thoroughly address the reviewers' reports. As you will see below, majority of reviewers are positive about the relevance and novelty of the study, yet some concerns have raised during revision. Reviewer 1 requests the inclusion of a schematic model, an evaluation of KD efficacy at the protein level, and an assessment of infected culture viability to better understand the functional consequences of icRNA sensing. Reviewer 2 notes that inflammasome activation appears extremely low compared to other stimuli and requests additional comparisons, including infection levels or IL-1B induction levels in uninfected cells, a quantitative WB of NLRP1 expression, a proximity ligation assay for interactions between dsRNA, the HIV-1 genome, and NLRP1, and an investigation into the fate of cells undergoing inflammasome activation. Reviewer 3 also suggests a proximity ligation assay. 

IMPORTANT: following discussions with the Academic Editor and the reviewers, we believe that further investigation into the inflammasome aspect could significantly strengthen the study. During cross-comments, Reviewer 2 proposed several informative experiments, such as flow cytometry to examine GSDMD pores on the cell surface, FLICA, PI uptake, and their combination with the GFP infection reporter gene. While these experiments are *not strictly required* for further consideration, we strongly encourage them, as they would substantially enhance the manuscript and increase its relevance to the field.

Given the extent of revision needed, we cannot make a decision about publication until we have seen the revised manuscript and your response to the reviewers' comments. Your revised manuscript is likely to be sent for further evaluation by all or a subset of the reviewers.

**IMPORTANT - SUBMITTING YOUR REVISION**

*Re-submission Checklist*

*Published Peer Review*

*PLOS Data Policy*

*Blot and Gel Data Policy*

Sincerely,

Melissa

Melissa Vazquez Hernandez, Ph.D.

Associate Editor

PLOS Biology

REVIEWERS' REPORTS:

Reviewer #1: 

An increasing amount of mechanisms for the detection of HIV RNAs are being reported and may contribute to chronic inflammation observed in PLWH. The authors previously reported that intron-containing HIV RNAs (icRNAs) can be sensed by MDA5/MAVS. Here, they report that in addition, cytosolic icRNAs can be sensed by macrophages and microglia to trigger secretion of IL-1ß. Through a series of conclusive experiments using inhibitors, gene silencing and virus variants, clear evidence is presented that this new mechanism involves NLRP1 and caspase-1 and is distinct from recently described mechanisms involving MDA5/MAVS or CARD8. This is a carefully conducted study that presents interesting and novel findings. The results are clear, for the most part generated using highly relevant primary target cells and the manuscript is very well written. Reflecting this maturity, this seems to be a revised manuscript version. I have not seen the initial submission and will focus on a few general points that could still be improved.

1) A schematic model that puts the newly discovered mechanisms in the context of previous reports would be helpful for the reader.

2) The efficacy of PRR knockdowns was assessed on the mRNA and functional level. Including an additional analysis on the protein level would be important.

3) Only little focus is put on the functional consequences of icRNA sensing (LDH release in 5F, 6J). A more detailed analysis of viability of infected cultures and assessing the impact of cell culture supernatants on the susceptibility of naïve cultures to infection would be a good addition. 

Reviewer #2: 

This is a novel and interesting pathway, and adds a new feature to our understanding of how NLRP1 can trigger inflammasome activation. Broadly, the experiments are well designed and executed, with various inhibitors of key steps of HIV-1 replication being exploited to carefully nail down the pathway driving the outcomes observed. With that said, I must note that the level of inflammasome activation induced via this pathway appears to be extremely low, especially in MDMs, leading to release of IL-1B levels that, while statistically significant and appear quite reproducible, do not reach the level of induction one sees for many other stimuli in many experiments. Ultimately, I am more interested in understanding how this sensing occurs in the well controlled and in many cases quite elegant experiments in the study, but I do think it is important to note the modest levels of inflammasome induction observed in these experiments. I leave it to the editor and reviewer to determine how this may impact the outcome of this review. 

There are also a few areas of weakness of the manuscript, described below, that should be addressed prior to publication, and a few technical or editorial issues that should also be addressed. 

Key issues: 

In figure S1, and as it related to figure 1, it would certainly be nice to appreciate the level of infection and/or IL-1B induction in cells with neither Vpx or dNs on board. These elements do not seem to be involved in figure 1, but the infection readout (p24) actually comes in higher than with Vpx or dNs treatment. Sine the authors make a point of saying dNs increase levels to the levels seen with Vpx, comparing each to an infection with neither, in the same experiment, seems important here. 

In figure S4, the authors use Immunofluorescence to compare the levels of NLRP1 in THP1s (transduced or not to express NLRP1), MDMs and iPSC derived microglia. Especially given the relatively modest level of response of MDMs, compared to microglia, and other studies which have shown low or undetectable levels of NLRP1 in some of these cell types, I think it is really critical to include a quantitative WB analysis of NLRP1 expression levels in all of these cell types. It appears that perhaps expression is a bit higher in iMGs compared to MDMs, which would ease my concerns related to the modest response in MDMs, as the authors do measure more impressive levels of IL-1� release from iMGs in figure 3. 

The mechanistic data in Figure 7 is quite interesting and provocative, and key to understanding how NLRP1 can act as described in the manuscript. While the authors note no colocalization of J2 antibody staining and NLRP1, a proximity ligation assay would be a much more sensitive and potentially informative indicator of association between dsRNA species, presumably parts of the HIV-1 genome with RNA duplexes, and NLRP1. 

The authors never really address the fate of cells undergoing NLRP1 mediated inflammasome activation, which is a weakness of the manuscript as submitted. Some LDH release data seems to suggest that cells die when this pathway is operative, but this does not seem to influence the number of infected cells in culture (as measured by GFP expression). Are only a very small subset of cells undergoing inflammasome activation via this pathway? Alternatively, are these cells avoiding pyroptotic cell death despite releasing IL-1B. Numerous approaches can be used to better understand this, including ASC spec formation or GSDMD cleavage and localization to the plasma membrane of infected cells. Does activation trigger PI uptake? Understanding the fate of infected cells following NLRP1 mediated inflammasome activation really seems critical to appreciate how this may influence HIV-1 pathogenesis/HAND or HIV-1 associated chronic inflammation, as suggested in the discussion. 

Minor Issues

Figure 1: 

In panel H, can the channels of the western blot be separated into capase-1 only and actin only panels. The point will not change, but the combination is not particularly aesthetically pleasing. Same comment for 2A

The generation of the delta Gag-Pol virus is rather poorly described. In the materials and methods, it is stated: Single-round viruses pseudotyped with VSV-G were generated from HEK293T cells via co-transfection of Lai∆envGFP proviral plasmids and VSV-G expression plasmid (pHCMV-G), and the packaging construct (psPAX2), where necessary. It would be worth clarifying which viruses were prepared with a packaging plasmid assist. Additionally, it might be worth a few lines of text in the result clarifying that this virus was made in 293Ts with a packaging plasmid, because it is not implausible that a reader without substantial expertise in HIV-1 molecular and cellular approaches would be very confused about how a delta Gag/Pol virus could be made or be infectious. Related: Line 206: suggesting that viral spread was not required for IL-1�…. I understand and agree with this interpretation, but if the authors could interpret it for the reader that might not understand the packaging plasmid dependence of this virus, it would make the outcome and interpretation more accessible to all readers. 

The experiments with M10/4x CTE virus are very nice, especially since they provide an alternative outcome than what is observed in the MAVS/IFN sensing pathway. One wonders if some of the old systems which measure the nuclear export of unspliced RNA might be able to trigger a response in these experiments if used in lentiviral vectors, although I wound not make this a requirement for publication. 

Line 76, note oddly formatted citations. Likely a formatting error, but if not, Im not clear on what this would mean. 

Line 369: Figure 4 is called out, should be figure 5. 

Reviewer #3 (Jeremy Luban): 

The literature has many reports of chronic inflammation, including inflammasome activation, in people living with HIV-1. Previously, the authors showed that unspliced, intron-containing HIV-1 RNA (icRNA) activates type 1 IFN in myeloid cells, including monocyte-derived macrophages (MDMs) and iPSC-derived microglia (iMGs). Here, they show that HIV-1 icRNA activates the inflammasome in these cells, as evidenced by IL-1B secretion and Caspase 1 cleavage. HIV-1 protease activity was not required for inflammasome activation. siRNAs targeting MAVS, STING, and UNC93B1 had no effect on HIV-1 icRNA-induced IL1B secretion. Previous work has shown that Rev-RRE-CRM1 is required for type 1 IFN induction by HIV-1 icRNA and that export of icRNA via a 4x-CTE would not substitute; here they show that the 4X-CTE is sufficient to activate IL1B. Knockdown of NLRP1 and CASPASE1, but not AIM2, NLRP3, CARD8, or caspase-4, decreased HIV-1 icRNA-induced IL-1β secretion and LDH release. THP1 cells do not produce NLRP1 protein and do not make IL1B in response to HIV-1 icRNA; when THP1 cells were transduced with an NLRP1-expressing lentivector, they did. 

This manuscript reports an important discovery: HIV-1 icRNA is detected by NLRP1 in myeloid cells and activates the inflammasome to secrete IL1B. The experiments are well controlled, the data convincing, and the manuscript is clear. Our suggestions are minor and can be quickly remedied. 

Page 4, line 81 mentions MDA5 without the relevant references. Refs 85 and 86 should be added here.

Page 5, line 103 suggests that IP10 data is shown in Fig 1. The authors could just indicate that this was previously shown (ref 56).

Fig 4F-J might be better split off as a separate figure that comes before Fig 4A-E.

Panels 4F-J are about host factors and would flow better into Fig 5, which is also about host factors. 

It is a little hard to distinguish the identities of the different bars in Fig 5B-F based on the color code. Perhaps the bars could be labeled as they were in panel A.

Figure 6: All figures prior to this one show statistical significance. Is IL1B induction by HIV-1 icRNA significantly above background?

Page 21, line 491: the authors state that dsRNA and NLRP1 colocalization was not detected. Would it be possible to detect colocalization using a proximity ligation assay? Such an experiment is not necessary but it would be fun to see the results.

---

## [Editor Report · Decision Letter 2]

17 Jun 2025

Dear Dr Gummuluru,

Thank you for your patience while we considered your revised manuscript "Expression of Intron-containing HIV-1 RNA Induces NLRP1 Inflammasome Activation in Myeloid Cells" for publication as a Research Article at PLOS Biology. This revised version of your manuscript has been evaluated by the PLOS Biology editors and the Academic Editor.

Based on our Academic Editor's assessment of your revision, we are likely to accept this manuscript for publication, provided you satisfactorily address the remaining editorial points. Please also make sure to address the following data and other policy-related requests.

a) In your Data Availability statement you say "The authors declare that the data that support the findings of this study are available within the paper and from the corresponding author upon reasonable request." Please be aware that we are an Open Science journal and all data and resources must be freely available wihtout limitation. This refer to your mention of "reasonable request". 

Please supply the numerical values either in the a supplementary file or as a permanent DOI’d deposition for the following figures:

Figure 1A-GI, 2B-G, 3C-F, 4AC-GI-L, 5AC-G, 6B-FHIJ, 7BDFGH, S1A-D, S2AD, S3AB, S5BCDEHIJ 

c) Please cite the location of the data clearly in all relevant main and supplementary Figure legends, e.g. “The data underlying this Figure can be found in S1 Data” or “The data underlying this Figure can be found in https://doi.org/10.5281/zenodo.XXXXX” 

d) We require the original, uncropped and minimally adjusted images supporting all blot and gel results reported in the Figures 1H, 2A, 4BH, 5B, 6AG, S2A, S5FG

We will require these files before a manuscript can be accepted so please prepare and upload them now. Please carefully read our guidelines for how to prepare and upload this data: https://journals.plos.org/plosbiology/s/figures#loc-blot-and-gel-reporting-requirements

e) For figures containing FACS data (Figures 3B, 4A, S2C, S5A), please provide the FCS files and a picture showing the successive plots and gates that were applied to the FCS files to generate the figure. We ask that you please deposit this data in the FlowRepository (https://flowrepository.org/) and provide the accession number/URL of the deposition in the Data Availability Statement in the online submission form.

f) Please ensure that your Data Statement in the submission system accurately describes where your data can be found and is in final format, as it will be published as written there.

g) Per journal policy, if you have generated any custom code during the course of this investigation, please make it available without restrictions upon publication. Please ensure that the code is sufficiently well documented and reusable, and that your Data Statement in the Editorial Manager submission system accurately describes where your code can be found.

We expect to receive your revised manuscript within two weeks. 

*Published Peer Review History*

*Press*

Sincerely,

Melissa

Melissa Vazquez Hernandez, Ph.D.

Associate Editor

PLOS Biology

---

## [Editor Report · Decision Letter 3]

15 Jul 2025

Dear Rahm,

Thank you for the submission of your revised Research Article "Expression of Intron-containing HIV-1 RNA Induces NLRP1 Inflammasome Activation in Myeloid Cells" for publication in PLOS Biology. On behalf of my colleagues and the Academic Editor, Frank Kirchhoff, I am pleased to say that we can in principle accept your manuscript for publication, provided you address any remaining formatting and reporting issues. These will be detailed in an email you should receive within 2-3 business days from our colleagues in the journal operations team; no action is required from you until then. Please note that we will not be able to formally accept your manuscript and schedule it for publication until you have completed any requested changes.

PRESS

Sincerely, 

Melissa

Melissa Vazquez Hernandez, Ph.D., Ph.D.

Associate Editor

PLOS Biology
